# Acceleration via Symplectic Discretization of High-Resolution Differential Equations

**Bin Shi**
University of California, Berkeley
binshi@berkeley.edu

**Simon S. Du**
Institute for Advanced Study
ssdu@ias.edu

**Weijie J. Su**
University of Pennsylvania
suw@wharton.upenn.edu

**Michael I. Jordan**
University of California, Berkeley
jordan@cs.berkeley.edu

## Abstract

We study first-order optimization algorithms obtained by discretizing ordinary differential equations (ODEs) corresponding to Nesterov's accelerated gradient methods (NAGs) and Polyak's heavy-ball method. We consider three discretization schemes: symplectic Euler **(S)**, explicit Euler **(E)** and implicit Euler **(I)** schemes. We show that the optimization algorithm generated by applying the symplectic scheme to a high-resolution ODE proposed by Shi et al. [2018] achieves the accelerated rate for minimizing both strongly convex functions and convex functions. On the other hand, the resulting algorithm either fails to achieve acceleration or is impractical when the scheme is implicit, the ODE is low-resolution, or the scheme is explicit.

## 1 Introduction

In this paper, we consider unconstrained minimization problems:

$$\min_{x \in \mathbb{R}^n} f(x), \tag{1.1}$$

where $f$ is a smooth convex function. The touchstone method in this setting is gradient descent (GD):

$$x_{k+1} = x_k - s\nabla f(x_k), \tag{1.2}$$

where $x_0$ is a given initial point and $s > 0$ is the step size. Whether there exist methods that improve on GD while remaining within the framework of first-order optimization is a subtle and important question.

Modern attempts to address this question date to Polyak [1964, 1987], who incorporated a momentum term into the gradient step, yielding a method that is referred to as the *heavy-ball method*:

$$y_{k+1} = x_k - s\nabla f(x_k), \quad x_{k+1} = y_{k+1} - \alpha(x_k - x_{k-1}), \tag{1.3}$$

where $\alpha > 0$ is a momentum coefficient. While the heavy-ball method provably attains a faster rate of *local* convergence than GD near a minimum of $f$, it generally does not provide a guarantee of acceleration *globally* [Polyak, 1964].

The next major development in first-order methods is due to Nesterov, who introduced first-order gradient methods that have a faster *global* convergence rate than GD [Nesterov, 1983, 2013]. For a $\mu$-strongly convex objective $f$ with $L$-Lipschitz gradients, Nesterov's *accelerated gradient method* (NAG-SC) involves the following pair of update equations:

$$y_{k+1} = x_k - s\nabla f(x_k), \quad x_{k+1} = y_{k+1} + \frac{1 - \sqrt{\mu s}}{1 + \sqrt{\mu s}} \left( y_{k+1} - y_k \right). \tag{1.4}$$

If one sets $s = 1/L$, then NAG-SC enjoys a $O\left((1 - \sqrt{\mu/L})^k\right)$ convergence rate, improving on the $O\left((1 - \mu/L)^k\right)$ convergence rate of GD. Nesterov also developed an accelerated algorithm (NAG-C) targeting smooth convex functions that are not strongly convex:

$$y_{k+1} = x_k - s\nabla f(x_k), \quad x_{k+1} = y_{k+1} + \frac{k}{k+3}(y_{k+1} - y_k). \tag{1.5}$$

This algorithm has a $O(L/k^2)$ convergence rate, which is faster than GD's $O(L/k)$ rate.

While yielding optimal and effective algorithms, the design principle of Nesterov's accelerated gradient algorithms (NAG) is not transparent. Convergence proofs for NAG often use the *estimate sequence* technique, which is inductive in nature and relies on series of algebraic tricks [Bubeck, 2015]. In recent years progress has been made in the understanding of acceleration by moving to a *continuous-time* formulation. In particular, Su et al. [2016] showed that as $s \to 0$, NAG-C converges to an ordinary differential equation (ODE) (Equation (2.2)); moreover, for this ODE, Su et al. [2016] derived a (continuous-time) convergence rate using a Lyapunov function, and further transformed this Lyapunov function to a discrete version and thereby provided a new proof of the fact that NAG-C enjoys a $O(L/k^2)$ rate.

Further progress in this vein has involved taking a variational point of view that derives ODEs from an underlying Lagrangian rather than from a limiting argument [Wibisono et al., 2016]. While this approach captures many of the variations of Nesterov acceleration presented in the literature, it does not distinguish between the heavy-ball dynamics and the NAG dynamics, and thus fails to distinguish between local and global acceleration. More recently, Shi et al. [2018] have returned to limiting arguments with a more sophisticated methodology. They have derived *high-resolution* ODEs for the heavy-ball method (Equation (2.4)), NAG-SC (Equation (2.5)) and NAG-C (Equation (2.6)). Notably, the high-resolution ODEs for the heavy-ball dynamics and the accelerated dynamics are different. Shi et al. [2018] also presented Lyapunov functions for these ODEs as well as the corresponding algorithms, and showed that these Lyapunov functions can be used to derive the accelerated rates of NAG-SC and NAG-C. A number of other papers have also contributed to the understanding of acceleration by working in a continuous-time formulation [Krichene and Bartlett, 2017, Krichene et al., 2015, Diakonikolas and Orecchia, 2017, Ghadimi and Lan, 2016, Diakonikolas and Orecchia, 2017].

This emerging literature has thus provided a new level of understanding of design principles for accelerated optimization. The design involves an interplay between continuous-time and discrete-time dynamics. ODEs are obtained either variationally or via a limiting scheme, and various properties of the ODEs are studied, including their convergence rate, topological aspects of their flow and their behavior under perturbation. Lyapunov functions play a key role in such analyses, and also allow aspects of the continuous-time analysis to be transferred to discrete time [see, e.g., Wilson et al., 2016].

And yet the literature has not yet provided a full exploration of the transition from continuous-time ODEs to discrete-time algorithms. Indeed, this transition is a non-trivial one, as evidenced by the decades of research on numerical methods for the discretization of ODEs, including most notably the sophisticated arsenal of techniques referred to as "geometric numerical integration" that are used for ODEs obtained from underlying variational principles [Hairer et al., 2006]. Recent work has begun to explore these issues; examples include the use of symplectic integrators by Betancourt et al. [2018] and the use of Runge-Kutta integration by Zhang et al. [2018]. However, these methods do not always yield proofs that accelerated rates are retained in discrete time, and when they do they involve implicit discretization, which is generally not practical except in the setting of quadratic objectives.

Thus we wish to address the following fundamental question:

*Can we systematically and provably obtain new accelerated methods via the numerical discretization of ordinary differential equations?*

Our approach to this question is a dynamical systems framework based on Lyapunov theory. Our main results are as follows:

1. In Section 3.1, we consider three simple numerical discretization schemes—symplectic Euler (**S**), explicit Euler (**E**) and implicit Euler (**I**) schemes—to discretize the high-resolution ODE of

Nesterov's accelerated method for strongly convex functions. We show that the optimization method generated by symplectic discretization achieves a $O((1 - O(1)\sqrt{\mu/L})^k)$ rate, thereby attaining acceleration. In sharp contrast, the implicit scheme is not practical for implementation, and the explicit scheme, while being simple, fails to achieve acceleration.

2. In Section 3.2, we apply these discretization schemes to the ODE for modeling the heavy-ball method, which can be viewed as a low-resolution ODE that lacks a gradient-correction term [Shi et al., 2018]. In contrast to the previous two cases of high-resolution ODEs, the symplectic scheme does not achieve acceleration for this low-resolution ODE. More broadly, in Appendix D we present more examples of low-resolution ODEs where symplectic discretization does *not* lead to acceleration.

3. Next, we apply the three simple Euler schemes to the high-resolution ODE of Nesterov's accelerated method for convex functions. Again, our Lyapunov analysis sheds light on the superiority of the symplectic scheme over the other two schemes. This is the subject of Section 4.

Taken together, the three findings have the implication that *high-resolution* ODEs and *symplectic* schemes are critical to achieving acceleration using numerical discretization. More precisely, in addition to allowing relatively simple implementations, symplectic schemes allow for a large step size without a loss of stability, in a manner akin to (but better than) implicit schemes. In stark contrast, in the setting of low-resolution ODEs, only the implicit schemes remain stable with a large step size, due to the lack of gradient correction. Moreover, the choice of Lyapunov function is equally essential to obtaining sharp convergence rates. This important fact is highlighted in Theorem A.6 in the Appendix, where we analyze GD by considering it as a discretization method for gradient flow (the ODE counterpart of GD). Using the discrete version of the Lyapunov function proposed in Su et al. [2016] instead of the classical one, we show that GD in fact minimizes the squared gradient norm (choosing the best iterate so far) at a rate of $O(L^2/k^2)$. Although this rate of convergence in the problem of squared gradient norm minimization is known in the literature [Nesterov, 2012], the Lyapunov function argument provides a systematic approach to obtaining this rate in this problem and others. In particular, this example demonstrates the usefulness and flexibility of Lyapunov functions as a mathematical tool for optimization problems.

## 2    Preliminaries

In this section, we introduce necessary notation, and review ODEs derived in previous work and three classical numerical discretization schemes.

We mostly follow the notation of Nesterov [2013], with slight modifications tailored to the present paper. Let $\mathcal{F}_L^1(\mathbb{R}^n)$ be the class of $L$-smooth convex functions defined on $\mathbb{R}^n$; that is, $f \in \mathcal{F}_L^1(\mathbb{R}^n)$ if $f(y) \geq f(x) + \langle \nabla f(x), y - x \rangle$ for all $x, y \in \mathbb{R}^n$ and its gradient is $L$-Lipschitz continuous in the sense that

$$\|\nabla f(x) - \nabla f(y)\| \leq L \|x - y\|,$$

where $\|\cdot\|$ denotes the standard Euclidean norm and $L > 0$ is the Lipschitz constant. The function class $\mathcal{F}_L^2(\mathbb{R}^n)$ is the subclass of $\mathcal{F}_L^1(\mathbb{R}^n)$ such that each $f$ has a Lipschitz-continuous Hessian. For $p = 1, 2$, let $\mathcal{S}_{\mu,L}^p(\mathbb{R}^n)$ denote the subclass of $\mathcal{F}_L^p(\mathbb{R}^n)$ such that each member $f$ is $\mu$-strongly convex for some $0 < \mu \leq L$. That is, $f \in \mathcal{S}_{\mu,L}^p(\mathbb{R}^n)$ if $f \in \mathcal{F}_L^p(\mathbb{R}^n)$ and $f(y) \geq f(x) + \langle \nabla f(x), y - x \rangle + \frac{\mu}{2} \|y - x\|^2$ for all $x, y \in \mathbb{R}^n$. Let $x^\star$ denote a minimizer of $f(x)$.

### 2.1    Approximating ODEs

In this section we list all of the ODEs that we will discretize in this paper. We refer readers to recent papers by Su et al. [2016], Wibisono et al. [2016] and Shi et al. [2018] for the rigorous derivations of these ODEs. We begin with the simplest. Taking the step size $s \to 0$ in Equation (1.2), we obtain the following ODE (gradient flow):

$$\dot{X} = -\nabla f(X), \tag{2.1}$$

with any initial $X(0) = x_0 \in \mathbb{R}^n$.

Next, by taking $s \to 0$ in Equation (1.5), Su et al. [2016] derived the low-resolution ODE of NAG-C:

$$\ddot{X} + \frac{3}{t}\dot{X} + \nabla f(X) = 0, \tag{2.2}$$

with $X(0) = x_0$ and $\dot{X}(0) = 0$. For strongly convex functions, by taking $s \to 0$, one can derive the following low-resolution ODE (see, for example, Wibisono et al. [2016])

$$\ddot{X} + 2\sqrt{\mu}\dot{X} + \nabla f(X) = 0 \tag{2.3}$$

that models both the heavy-ball method and NAG-SC. This ODE has the same initial conditions as (2.2).

Recently, Shi et al. [2018] proposed high-resolution ODEs for modeling acceleration methods. The key ingredient in these ODEs is that the $O(\sqrt{s})$ terms are preserved in the ODEs. As a result, the heavy-ball method and NAG-SC have different models as ODEs.

(a) If $f \in \mathcal{S}_{\mu,L}^1(\mathbb{R}^n)$, the high-resolution ODE of the heavy-ball method (1.3) is

$$\ddot{X} + 2\sqrt{\mu}\dot{X} + (1 + \sqrt{\mu s})\nabla f(X) = 0, \tag{2.4}$$

with $X(0) = x_0$ and $\dot{X}(0) = -\frac{2\sqrt{s}\nabla f(x_0)}{1+\sqrt{\mu s}}$. This ODE has essentially the same properties as its low-resolution counterpart (2.3) due to the absence of $\nabla^2 f(X)\dot{X}$.

(b) If $f \in \mathcal{S}_{\mu,L}^2(\mathbb{R}^n)$, the high-resolution ODE of NAG-SC (1.4) is

$$\ddot{X} + 2\sqrt{\mu}\dot{X} + \sqrt{s}\nabla^2 f(X)\dot{X} + (1 + \sqrt{\mu s})\nabla f(X) = 0, \tag{2.5}$$

with $X(0) = x_0$ and $\dot{X}(0) = -\frac{2\sqrt{s}\nabla f(x_0)}{1+\sqrt{\mu s}}$.

(c) If $f \in \mathcal{F}_L^2(\mathbb{R}^n)$, the high-resolution ODE of NAG-C (1.5) is

$$\ddot{X} + \frac{3}{t}\dot{X} + \sqrt{s}\nabla^2 f(X)\dot{X} + \left(1 + \frac{3\sqrt{s}}{2t}\right)\nabla f(X) = 0 \tag{2.6}$$

for $t \geq 3\sqrt{s}/2$, with $X(3\sqrt{s}/2) = x_0$ and $\dot{X}(3\sqrt{s}/2) = -\sqrt{s}\nabla f(x_0)$.

## 2.2 Discretization schemes

To discretize ODEs (2.1)-(2.6), we replace $\dot{X}$ by $x_{k+1} - x_k$, $\dot{V}$ by $v_{k+1} - v_k$ and replace other terms with approximations. Different discretization schemes correspond to different approximations.

- The most straightforward scheme is the explicit scheme, which uses the following approximation rule:
$$x_{k+1} - x_k = \sqrt{s}v_k, \qquad \sqrt{s}\nabla^2 f(x_k)v_k \approx \nabla f(x_{k+1}) - \nabla f(x_k).$$

- Another discretization scheme is the implicit scheme, which uses the following approximation rule:
$$x_{k+1} - x_k = \sqrt{s}v_{k+1}, \qquad \sqrt{s}\nabla^2 f(x_{k+1})v_{k+1} \approx \nabla f(x_{k+1}) - \nabla f(x_k).$$
Note that compared with the explicit scheme, the implicit scheme is not practical because the update of $x_{k+1}$ requires knowing $v_{k+1}$ while the update of $v_{k+1}$ requires knowing $x_{k+1}$.

- The last discretization scheme considered in this paper is the symplectic scheme, which uses the following approximation rule.
$$x_{k+1} - x_k = \sqrt{s}v_k, \qquad \sqrt{s}\nabla^2 f(x_{k+1})v_k \approx \nabla f(x_{k+1}) - \nabla f(x_k).$$
Note this scheme is practical because the update of $x_{k+1}$ only requires knowing $v_k$.

We remark that for low-resolution ODEs, there is no $\nabla^2 f(x)$ term, whereas for high-resolution ODEs, we have this term and we use the difference of gradients to approximate this term. This additional approximation term is critical to acceleration.

# 3 High-Resolution ODEs for Strongly Convex Functions

This section considers numerical discretization of the high-resolution ODEs of NAG-SC and the heavy-ball method using the symplectic Euler, explicit Euler and implicit Euler scheme. In particular, we compare rates of convergence towards the objective minimum of the three simple Euler schemes and the two methods (NAG-SC and the heavy-ball method) in Section 3.1 and Section 3.2, respectively. For both cases, the associated symplectic scheme is shown to exhibit surprisingly similarity to the corresponding classical method.

### 3.1 NAG-SC

The high-resolution ODE (2.5) of NAG-SC can be equivalently written in the phase space as

$$\dot{X} = V, \qquad \dot{V} = -2\sqrt{\mu}V - \sqrt{s}\nabla^2 f(X)V - (1 + \sqrt{\mu s})\nabla f(X), \qquad (3.1)$$

with the initial conditions $X(0) = x_0$ and $V(0) = -\frac{2\sqrt{s}\nabla f(x_0)}{1+\sqrt{\mu s}}$. For any $f \in \mathcal{S}_{\mu,L}^2(\mathbb{R}^n)$, Theorem 1 of Shi et al. [2018] shows that the solution $X = X(t)$ of the ODE (2.5) satisfies

$$f(X) - f(x^\star) \le \frac{2\left\| x_0 - x^\star \right\|^2}{s} e^{-\frac{\sqrt{\mu} t}{4}},$$

for any step size $0 < s \le 1/L$. In particular, setting the step size to $s = 1/L$, we get

$$f(X) - f(x^\star) \le 2L \left\| x_0 - x^\star \right\|^2 e^{-\frac{\sqrt{\mu} t}{4}}.$$

In the phase space representation, NAG-SC is formulated as

$$\begin{cases} x_{k+1} - x_k = \sqrt{s}v_k \\ v_{k+1} - v_k = -\dfrac{2\sqrt{\mu s}}{1 - \sqrt{\mu s}}v_{k+1} - \sqrt{s}(\nabla f(x_{k+1}) - \nabla f(x_k)) - \dfrac{1 + \sqrt{\mu s}}{1 - \sqrt{\mu s}} \cdot \sqrt{s}\nabla f(x_{k+1}), \end{cases}$$
$$(3.2)$$

with the initial condition $v_0 = -\frac{2\sqrt{s}\nabla f(x_0)}{1+\sqrt{\mu s}}$ for any $x_0$. This method maintains the accelerated rate of the ODE by recognizing

$$f(x_k) - f(x^\star) \le \frac{5L \left\| x_0 - x^\star \right\|^2}{(1 + \sqrt{\mu/L}/12)^k};$$

(see Theorem 3 in Shi et al. [2018]) and the identification $t \approx k\sqrt{s}$.

Viewing NAG-SC as a numerical discretization of (2.5), one might wonder if any of the three simple Euler schemes—symplectic Euler scheme, explicit Euler scheme, and implicit Euler scheme—maintain the accelerated rate in discretizing the high-resolution ODE. For clarity, the update rules of the three schemes are given as follows, each with the initial points $x_0$ and $v_0 = -\frac{2\sqrt{s}\nabla f(x_0)}{1+\sqrt{\mu s}}$.

**Euler scheme of** (3.1)**: (S), (E) and (I) respectively**

(S) $\begin{cases} x_{k+1} - x_k = \sqrt{s}v_k \\ v_{k+1} - v_k = -2\sqrt{\mu s}v_{k+1} - \sqrt{s}\left(\nabla f(x_{k+1}) - \nabla f(x_k)\right) - \sqrt{s}(1 + \sqrt{\mu s})\nabla f(x_{k+1}). \end{cases}$

(E) $\begin{cases} x_{k+1} - x_k = \sqrt{s}v_k \\ v_{k+1} - v_k = -2\sqrt{\mu s}v_k - \sqrt{s}\left(\nabla f(x_{k+1}) - \nabla f(x_k)\right) - \sqrt{s}(1 + \sqrt{\mu s})\nabla f(x_k). \end{cases}$

(I) $\begin{cases} x_{k+1} - x_k = \sqrt{s}v_{k+1} \\ v_{k+1} - v_k = -2\sqrt{\mu s}v_{k+1} - \sqrt{s}\left(\nabla f(x_{k+1}) - \nabla f(x_k)\right) - \sqrt{s}(1 + \sqrt{\mu s})\nabla f(x_{k+1}). \end{cases}$

Among the three Euler schemes, the symplectic scheme is the *closest* to NAG-SC (3.2). More precisely, NAG-SC differs from the symplectic scheme only in an additional factor of $\frac{1}{1-\sqrt{\mu s}}$ in the second line of (3.2). When the step size $s$ is small, NAG-SC is, roughly speaking, a symplectic method if we make use of $\frac{1}{1-\sqrt{\mu s}} \approx 1$. In relating to the literature, the connection between accelerated methods and the symplectic schemes has been explored in Betancourt et al. [2018], which mainly considers the leapfrog integrator, a second-order symplectic integrator. In contrast, the symplectic Euler scheme studied in this paper is a first-order symplectic integrator.

Interestingly, the close resemblance between the two algorithms is found not only in their formulations, but also in their convergence rates, which are *both* accelerated as shown by Theorem B.1 and Theorem 3.1.

Note that the discrete Lyapunov function used in the proof of the symplectic Euler scheme of (3.1) is

$$\mathcal{E}(k) = \frac{1}{4}\left\| v_k \right\|^2 + \frac{1}{4}\left\| 2\sqrt{\mu}(x_{k+1} - x^\star) + v_k + \sqrt{s}\nabla f(x_k) \right\|^2$$

$$+ \left(1 + \sqrt{\mu s}\right)\left(f(x_k) - f(x^\star)\right) - \frac{\left(1 + \sqrt{\mu s}\right)^2}{1 + 2\sqrt{\mu s}} \cdot \frac{s}{2} \left\| \nabla f(x_k) \right\|^2 . \tag{3.3}$$

The proof of Theorem B.1 is deferred to Appendix B.1. The following result is a useful consequence of this theorem.

**Theorem 3.1** (Discretization of NAG-SC ODE). For any $f \in \mathcal{S}_{\mu,L}^1(\mathbb{R}^n)$, the following conclusions hold:

(a) Taking step size $s = 4/(9L)$, the symplectic Euler scheme of (3.1) satisfies

$$f(x_k) - f(x^\star) \leq \frac{5L \left\| x_0 - x^\star \right\|^2}{\left(1 + \frac{1}{9}\sqrt{\frac{\mu}{L}}\right)^k} . \tag{3.4}$$

(b) Taking step size $s = \mu/(100L^2)$, the explicit Euler scheme of (3.1) satisfies

$$f(x_k) - f(x^\star) \leq 3L \left\| x_0 - x^\star \right\|^2 \left(1 - \frac{\mu}{80L}\right)^k . \tag{3.5}$$

(c) Taking step size $s = 1/L$, the implicit Euler scheme of (3.1) satisfies

$$f(x_k) - f(x^\star) \leq \frac{13 \left\| x_0 - x^\star \right\|^2}{4\left(1 + \frac{1}{4}\sqrt{\frac{\mu}{L}}\right)^k} . \tag{3.6}$$

In addition, Theorem 3.1 shows that the implicit scheme also achieves acceleration. However, unlike NAG-SC, the symplectic scheme, and the explicit scheme, the implicit scheme is generally not easy to use in practice because it requires solving a nonlinear fixed-point equation when the objective is not quadratic. On the other hand, the explicit scheme can only take a smaller step size $O(\mu/L^2)$, which prevents this scheme from achieving acceleration.

## 3.2 The heavy-ball method

We turn to the heavy-ball method ODE (2.4), whose phase space representation reads

$$\dot{X} = V, \qquad \dot{V} = -2\sqrt{\mu}V - (1 + \sqrt{\mu s})\nabla f(X), \tag{3.7}$$

with the initial conditions $X(0) = x_0$ and $V(0) = -\frac{2\sqrt{s}\nabla f(x_0)}{1 + \sqrt{\mu s}}$. Theorem 2 in Shi et al. [2018] shows that the solution $X = X(t)$ to this ODE satisfies

$$f(X(t)) - f(x^\star) \leq \frac{7 \left\| x_0 - x^\star \right\|^2}{2s} e^{-\frac{\sqrt{\mu}t}{4}} ,$$

for $f \in \mathcal{S}_{\mu,L}^1(\mathbb{R}^n)$ and any step size $0 < s \leq 1/L$. In particular, taking $s = 1/L$ gives

$$f(X(t)) - f(x^\star) \leq \frac{7L \left\| x_0 - x^\star \right\|^2}{2} e^{-\frac{\sqrt{\mu}t}{4}} .$$

Returning to the discrete regime, Polyak's heavy-ball method uses the following update rule:

$$\begin{cases} x_{k+1} - x_k = \sqrt{s}v_k \\ v_{k+1} - v_k = -\frac{2\sqrt{\mu s}}{1 - \sqrt{\mu s}}v_{k+1} - \frac{1 + \sqrt{\mu s}}{1 - \sqrt{\mu s}} \cdot \sqrt{s}\nabla f(x_{k+1}), \end{cases}$$

which attains a non-accelerated rate (see Theorem 4 of Shi et al. [2018]):

$$f(x_k) - f(x^\star) \leq \frac{5L \left\| x_0 - x^\star \right\|^2}{\left(1 + \frac{\mu}{16L}\right)^k} . \tag{3.8}$$

The three simple Euler schemes for numerically solving the ODE (2.4) are given as follows. Every scheme starts with any arbitrary $x_0$ and $v_0 = -\frac{2\sqrt{s}\nabla f(x_0)}{1 + \sqrt{\mu s}}$. As in the case of NAG-SC, the symplectic scheme is the closest to the heavy-ball method.

**Euler scheme of** (3.7)**: (S), (E) and (I) respectively**

(S)
$$\begin{cases} x_{k+1} - x_k = \sqrt{s}v_k, \\ v_{k+1} - v_k = -2\sqrt{\mu s}v_{k+1} - \sqrt{s}(1 + \sqrt{\mu s})\nabla f(x_{k+1}). \end{cases}$$

(E)
$$\begin{cases} x_{k+1} - x_k = \sqrt{s}v_k \\ v_{k+1} - v_k = -2\sqrt{\mu s}v_k - \sqrt{s}(1 + \sqrt{\mu s})\nabla f(x_k). \end{cases}$$

(I)
$$\begin{cases} x_{k+1} - x_k = \sqrt{s}v_{k+1} \\ v_{k+1} - v_k = -2\sqrt{\mu s}v_{k+1} - \sqrt{s}(1 + \sqrt{\mu s})\nabla f(x_{k+1}). \end{cases}$$

The theorem below characterizes the convergence rates of the three schemes. This theorem is extended to general step sizes by Theorem B.2 in Appendix B.2.

**Theorem 3.2** (Discretization of heavy-ball ODE). For any $f \in \mathcal{S}_{\mu,L}^1(\mathbb{R}^n)$, the following conclusions hold:

(a) Taking step size $s = \mu/(16L^2)$, the symplectic Euler scheme of (3.7) satisfies

$$f(x_k) - f(x^\star) \le \frac{3L \|x_0 - x^\star\|^2}{\left(1 + \frac{\mu}{16L}\right)^k}. \tag{3.9}$$

(b) Taking step size $s = \mu/(36L^2)$, the explicit Euler scheme of (3.7) satisfies

$$f(x_k) - f(x^\star) \le 3L \|x_0 - x^\star\|^2 \left(1 - \frac{\mu}{48L}\right)^k. \tag{3.10}$$

(c) Taking step size $s = 1/L$, the implicit Euler scheme of (3.7) satisfies

$$f(x_k) - f(x^\star) \le \frac{15L \|x_0 - x^\star\|^2}{4\left(1 + \frac{1}{4}\sqrt{\frac{\mu}{L}}\right)^k}. \tag{3.11}$$

Taken together, (3.8) and Theorem 3.2 imply that neither the heavy-ball method nor the symplectic scheme attains an accelerated rate. In contrast, the implicit scheme achieves acceleration as in the NAG-SC case, but it is impractical except for quadratic objectives.

## 4 High-Resolution ODEs for Convex Functions

In this section, we turn to numerical discretization of the high-resolution ODE (2.6) related to NAG-C. All proofs are deferred to Appendix C. This ODE in the phase space representation reads [Shi et al., 2018] as follows:

$$\dot{X} = V, \quad \dot{V} = -\frac{3}{t} \cdot V - \sqrt{s}\nabla^2 f(X)V - \left(1 + \frac{3\sqrt{s}}{2t}\right)\nabla f(X), \tag{4.1}$$

with $X(3\sqrt{s}/2) = x_0$ and $V(3\sqrt{s}/2) = -\sqrt{s}\nabla f(x_0)$. Theorem 5 of Shi et al. [2018] shows that Let $f \in \mathcal{F}_L^1(\mathbb{R}^n)$. For any step size $0 < s \le 1/L$, the solution $X = X(t)$ of the high-resolution ODE (2.6) satisfies

$$\begin{cases} f(X) - f(x^\star) \le \dfrac{(4 + 3sL) \|x_0 - x^\star\|^2}{t(2t + \sqrt{s})} \\ \inf\limits_{t_0 \le u \le t} \|\nabla f(X(u))\|^2 \le \dfrac{(12 + 9sL) \|x_0 - x^\star\|^2}{2\sqrt{s}(t^3 - t_0^3)} \end{cases}, \tag{4.2}$$

for any $t > t_0 = 1.5\sqrt{s}$. A caveat here is that it is unclear how to use a Lyapunov function to prove convergence of the (simple) explicit, symplectic or implicit Euler scheme by direct numerical discretization of the ODE (2.2). See Appendix C.2 for more discussion on this point. Therefore, we slightly modify the ODE to the following one:

$$\dot{X} = V, \quad \dot{V} = -\frac{3}{t} \cdot V - \sqrt{s}\nabla^2 f(X)V - \left(1 + \frac{3\sqrt{s}}{t}\right)\nabla f(X). \tag{4.3}$$

The only difference is in the third term on the right-hand side of the second equation, where we replace $\left(1 + \frac{3\sqrt{s}}{2t}\right)\nabla f(X)$ by $\left(1 + \frac{3\sqrt{s}}{t}\right)\nabla f(X)$. Now, we apply the three schemes on this (modified) ODE in the phase space, including the original NAG-C, which all start with $x_0$ and $v_0 = -\sqrt{s}\nabla f(x_0)$.

**Euler scheme of** (4.3)**: (S), (E) and (I) respectively**

**(S)**
$$\begin{cases} x_{k+1} - x_k = \sqrt{s}v_k \\ v_{k+1} - v_k = -\dfrac{3}{k+1}v_{k+1} - \sqrt{s}\left(\nabla f(x_{k+1}) - \nabla f(x_k)\right) - \sqrt{s}\left(\dfrac{k+4}{k+1}\right)\nabla f(x_{k+1}). \end{cases}$$

**(E)**
$$\begin{cases} x_{k+1} - x_k = \sqrt{s}v_k \\ v_{k+1} - v_k = -\dfrac{3}{k}v_k - \sqrt{s}\left(\nabla f(x_{k+1}) - \nabla f(x_k)\right) - \sqrt{s}\left(\dfrac{k+3}{k}\right)\nabla f(x_k). \end{cases}$$

**(I)**
$$\begin{cases} x_{k+1} - x_k = \sqrt{s}v_{k+1} \\ v_{k+1} - v_k = -\dfrac{3}{k+1}v_{k+1} - \sqrt{s}\left(\nabla f(x_{k+1}) - \nabla f(x_k)\right) - \sqrt{s}\left(\dfrac{k+4}{k+1}\right)\nabla f(x_{k+1}). \end{cases}$$

**Theorem 4.1.** Let $f \in \mathcal{F}_L^1(\mathbb{R}^n)$. The following statements are true:

(a) For any step size $0 < s \le 1/(3L)$, the symplectic Euler scheme of (4.3) (original NAG-C) satisfies

$$f(x_k) - f(x^\star) \le \frac{119\|x_0 - x^\star\|^2}{s(k+1)^2}, \quad \min_{0 \le i \le k}\|\nabla f(x_i)\|^2 \le \frac{8568\|x_0 - x^\star\|^2}{s^2(k+1)^3}; \quad (4.4)$$

(b) Taking any step size $0 < s \le 1/L$, the implicit Euler scheme of (4.3) satisfies

$$f(x_k) - f(x^\star) \le \frac{(3sL+2)\|x_0 - x^\star\|^2}{s(k+2)(k+3)}, \quad \min_{0 \le i \le k}\|\nabla f(x_i)\|^2 \le \frac{(3sL+2)\|x_0 - x^\star\|^2}{s^2(k+1)^3}. \quad (4.5)$$

Note that Theorem 4.1 (a) is the same as Theorem 6 of Shi et al. [2018]. The explicit Euler scheme does not guarantee convergence; see the analysis in Appendix C.1.

## 5 Discussion

In this paper, we have analyzed the convergence rates of three numerical discretization schemes—the symplectic Euler scheme, explicit Euler scheme, and implicit Euler scheme—applied to ODEs that are used for modeling Nesterov's accelerated methods and Polyak's heavy-ball method. The symplectic scheme is shown to achieve accelerated rates for the high-resolution ODEs of NAG-SC and (slightly modified) NAG-C [Shi et al., 2018], whereas no acceleration rates are observed when the same scheme is used to discretize the low-resolution counterparts [Su et al., 2016]. For comparison, the explicit scheme only allows for a small step size in discretizing these ODEs in order to ensure stability, thereby failing to achieve acceleration. Although the implicit scheme is proved to yield accelerated methods no matter whether high-resolution or low-resolution ODEs are discretized, this scheme is generally not practical except for a limited number of cases (for example, quadratic objectives).

We conclude this paper by presenting several directions for future work. This work suggests that both symplectic schemes and high-resolution ODEs are crucial for numerical discretization to achieve acceleration. It would be of interest to formalize and prove this assertion. For example, does any higher-order symplectic scheme maintain acceleration for the high-resolution ODEs of NAGs? What is the fundamental mechanism of the gradient correction in high-resolution ODE in stabilizing symplectic discretization? Moreover, since the discretizations are applied to the modified high-resolution ODE of NAG-C, it is tempting to perform a comparison study between the two high-resolution ODEs in terms of discretization properties. Finally, recognizing Nesterov's method (NAG-SC) is very similar to, but still different from, the corresponding symplectic scheme, one can design new algorithms as interpolations of the two methods; it would be interesting to investigate the convergence properties of these new algorithms.

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
