[Supplementary Material]

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

# Contents

# A   Gradient Flow

## A.1   Convergence rate of gradient flow

The following theorem is the continuous-time version of Theorem 2.1.15 in Nesterov [2013].

**Theorem A.1.** Let $f \in \mathcal{S}_{\mu,L}^1(\mathbb{R}^n)$. The solution $X = X(t)$ to the gradient flow (2.1) satisfies

$$\|X - x^\star\| \le \mathrm{e}^{-\mu t}\|x_0 - x^\star\|.$$

*Proof.* Taking the following Lyapunov function

$$\mathcal{E} = \|X - x^\star\|^2,$$

327 we calculate its time derivative as

$$\frac{\mathrm{d}\mathcal{E}}{\mathrm{d}t} = 2\left\langle \dot{X}, X - x^\star \right\rangle$$
$$= -2\left\langle \nabla f(X), X - x^\star \right\rangle$$
$$\leq -2\mu \left\| X - x^\star \right\|^2.$$

328 Thus, we complete the proof. □

329 The theorem below is a continuous version of Theorem 2.1.14 in Nesterov [2013].

330 **Theorem A.2.** Let $f \in \mathcal{F}_L^1(\mathbb{R}^n)$. The solution $X = X(t)$ to the gradient flow (2.1) satisfies

$$f(X) - f(x^\star) \leq \frac{(f(x_0) - f(x^\star)) \left\| x_0 - x^\star \right\|^2}{t(f(x_0) - f(x^\star)) + \left\| x_0 - x^\star \right\|^2}.$$

331 *Proof.* The time derivative of the distance function is

$$\frac{\mathrm{d}}{\mathrm{d}t} \left\| X - x^\star \right\|^2 = 2\left\langle \dot{X}, X - x^\star \right\rangle$$
$$= -2\left\langle \nabla f(X), X - x^\star \right\rangle$$
$$\leq 0.$$

332 We define a Lyapunov function as

$$\mathcal{E} = f(X) - f(x^\star).$$

333 With the basic convex inequality for $f \in \mathcal{F}_L^1(\mathbb{R}^n)$, we have

$$f(X) - f(x^\star) \leq \left\langle \nabla f(X), X - x^\star \right\rangle \leq \left\| \nabla f(X) \right\| \left\| x_0 - x^\star \right\|.$$

334 Furthermore, we obtain that the time derivative is

$$\frac{\mathrm{d}\mathcal{E}}{\mathrm{d}t} = \left\langle \nabla f(X), \dot{X} \right\rangle = -\left\| \nabla f(X) \right\|^2 \leq -\frac{(f(X) - f(x^\star))^2}{\left\| x_0 - x^\star \right\|^2} = -\frac{\mathcal{E}^2}{\left\| x_0 - x^\star \right\|^2}.$$

335 Hence, the convergence rate is

$$f(X) - f(x^\star) \leq \frac{(f(x_0) - f(x^\star)) \left\| x_0 - x^\star \right\|^2}{t(f(x_0) - f(x^\star)) + \left\| x_0 - x^\star \right\|^2}.$$

336 □

337 The following theorem is based on the Lyapunov function for gradient flow (2.1) in Su et al. [2016].

338 **Theorem A.3.** Let $f \in \mathcal{F}_L^1(\mathbb{R}^n)$. The solution $X = X(t)$ to the gradient flow (2.1) satisfies

$$\begin{cases} f(X) - f(x^\star) \leq \dfrac{\left\| x_0 - x^\star \right\|^2}{2t} \\ \min\limits_{0 \leq u \leq t} \left\| \nabla f(X(u)) \right\|^2 \leq \dfrac{\left\| x_0 - x^\star \right\|^2}{t^2}. \end{cases}$$

339 *Proof.* The Lyapunov function is

$$\mathcal{E} = t(f(X) - f(x^\star)) + \frac{1}{2} \left\| X - x^\star \right\|^2.$$

340 We calculate its time derivative as

$$\frac{\mathrm{d}\mathcal{E}}{\mathrm{d}t} = f(X) - f(x^\star) + t\left\langle \nabla f(X), \dot{X} \right\rangle + \left\langle X - x^\star, \dot{X} \right\rangle$$
$$= f(X) - f(x^\star) - \left\langle \nabla f(X), X - x^\star \right\rangle - t\left\| \nabla f(X) \right\|_2^2$$
$$\leq -t\left\| \nabla f(X) \right\|^2.$$

341 Thus, we complete the proof. □

**Remark A.1.** From the view of Lyapunov function, Theorem A.3 is essentially different from Theorem A.2. When the Lyapunov function

$$\mathcal{E} = t\left(f(X) - f(x^\star)\right) + \frac{1}{2}\left\|X - x^\star\right\|^2$$

is used to take place of that

$$\mathcal{E} = f(X) - f(x^\star),$$

the same convergence rate for function value is not only obtained by the simple way of calculation, but we can also capture an advanced faster speed of the squared gradient norm. From this view, constructing Lyapunov function is a more powerful and advanced mathematical tool for optimization.

## A.2  Explicit Euler scheme

The corresponding explicit-scheme version of Theorem A.1 is just Theorem 2.1.15 in Nesterov [2013]. We state it below.

**Theorem A.4** (Theorem 2.1.15, Nesterov [2013]). Let $f \in \mathcal{S}_{\mu,L}^1(\mathbb{R}^n)$. Taking any step size $0 < s \leq 2/(\mu + L)$, the iterates $\{x_k\}_{k=0}^\infty$ generated by GD (1.2) satisfy

$$\left\|x_k - x^\star\right\|^2 \leq \left(1 - \frac{2\mu L s}{\mu + L}\right)\left\|x_0 - x^\star\right\|^2.$$

In addition, if the step size is set to $s = 2/(\mu + L)$, we get

$$\left\|x_k - x^\star\right\|^2 \leq \left(\frac{L - \mu}{L + \mu}\right)^2 \left\|x_0 - x^\star\right\|^2.$$

This proof is from Nesterov [2013]. The only conceptual difference is that we use the Lyapunov function

$$\mathcal{E}(k) = \left\|x_k - x^\star\right\|^2,$$

instead of the distance function $r_k$ in Nesterov [2013].

The corresponding explicit version of Theorem A.2 is Theorem 2.1.14 in Nesterov [2013]. We also state it as follows.

**Theorem A.5** (Theorem 2.1.14, Nesterov [2013]). Let $f \in \mathcal{F}_L^1(\mathbb{R}^n)$. Taking any step size $0 < s < 2/L$, the iterates $\{x_k\}_{k=0}^\infty$ generated by GD (1.2) satisfy

$$f(x_k) - f(x^\star) \leq \frac{2\left(f(x_0) - f(x^\star)\right)\left\|x_0 - x^\star\right\|^2}{2\left\|x_0 - x^\star\right\|^2 + ks(2 - Ls)\left(f(x_0) - f(x^\star)\right)}. \tag{A.1}$$

In addition, if the step size is set to $s = 1/L$, we get

$$f(x_k) - f(x^\star) \leq \frac{2L\left\|x_0 - x^\star\right\|^2}{k + 4}. \tag{A.2}$$

Again, Nesterov [2013] uses the Lyapunov function $\mathcal{E}(k)$ instead of $r_k$.

Finally, we show the corresponding discrete version of Theorem A.3, highlighting the ODE-based approach and the importance of Lyapunov functions in proofs.

**Theorem A.6.** Let $f \in \mathcal{F}_L^1(\mathbb{R}^n)$. Taking any step size $0 < s \leq 1/L$, the iterates $\{x_k\}_{k=0}^\infty$ generated by GD (1.2) satisfy

$$\begin{cases} f(x_k) - f(x^\star) \leq \dfrac{\left\|x_0 - x^\star\right\|^2}{2ks} \\[2mm] \min\limits_{0 \leq i \leq k} \left\|\nabla f(x_i)\right\|^2 \leq \dfrac{2\left\|x_0 - x^\star\right\|^2}{s^2(k+1)(k+2)}. \end{cases} \tag{A.3}$$

In addition, if the step size is set $s = 1/L$, we have

$$\begin{cases} f(x_k) - f(x^\star) \leq \dfrac{L\left\|x_0 - x^\star\right\|^2}{2k} \\[2mm] \min\limits_{0 \leq i \leq k} \left\|\nabla f(x_i)\right\|^2 \leq \dfrac{2L^2\left\|x_0 - x^\star\right\|^2}{(k+1)(k+2)}. \end{cases} \tag{A.4}$$

368 To obtain this result, we use a Lyapunov function that is different from the standard analysis of
369 gradient descent, which uses the Lyapunov function $\mathcal{E}(k) \triangleq f(x_k) - f(x^\star)$. This Lyapunov
370 function yields the $O(L/k)$ convergence rate for the function value. For the squared gradient norm,
371 however, this Lyapunov function can only exploit the $L$-smoothness property that transforms the
372 function value to the gradient norm, giving the sub-optimal $O(L^2/k)$ rate, due to the absence
373 of gradient information in this function. Our proof uses a different Lyapunov function: $\mathcal{E}(k) =$
374 $ks\left(f(x_k) - f(x^\star)\right) + \frac{1}{2}\left\|x_k - x^\star\right\|^2$.

375 *Proof.* The corresponding discrete Lyapunov function is constructed as below

$$\mathcal{E}(k) = ks\left(f(x_k) - f(x^\star)\right) + \frac{1}{2}\left\|x_k - x^\star\right\|^2,$$

376 from which we get

$$
\begin{aligned}
&\mathcal{E}(k+1) - \mathcal{E}(k) \\
=& s\left(f(x_k) - f(x^\star)\right) + (k+1)s\left(f(x_{k+1}) - f(x_k)\right) + \frac{1}{2}\left\langle x_{k+1} - x_k, x_{k+1} + x_k - 2x^\star\right\rangle \\
\leq& s\left(f(x_k) - f(x^\star) - \left\langle \nabla f(x_k), x_k - x^\star\right\rangle\right) + (k+1)s\left\langle \nabla f(x_k), x_{k+1} - x_k\right\rangle \\
&+ \left[\frac{(k+1)sL}{2} + \frac{1}{2}\right]\left\|x_{k+1} - x_k\right\|^2 \\
\leq& s^2\left[-\frac{1}{2Ls} - (k+1) + \frac{(k+1)sL}{2} + \frac{1}{2}\right]\left\|\nabla f(x_k)\right\|^2 \\
\leq& -\frac{s^2}{2}(k+1)\left\|\nabla f(x_k)\right\|^2
\end{aligned}
$$

377 Taking $k_0$ in the assumption completes the proof. $\qquad\square$

378 **Remark A.2.** Same as the continuous ODE in Remark A.1, from view of the discrete algorithm, we
379 can find the apunov function is a more powerful and advanced mathematical tool.

380 **A.3 Implicit Euler scheme**

381 Next, we consider the implicit Euler scheme of the gradient flow (2.1) as

$$x_{k+1} = x_k - s\nabla f(x_{k+1}), \tag{A.5}$$

382 with any initial $x_0 \in \mathbb{R}^n$. The corresponding implicit version of Theorem A.1 is shown as below.

383 **Theorem A.7.** Let $f \in \mathcal{S}_{\mu,L}^1(\mathbb{R}^n)$, the iterates $\{x_k\}_{k=0}^\infty$ generated by implicit gradient descent (A.5)
384 satisfy

$$\left\|x_k - x^\star\right\| \leq \frac{1}{(1+\mu s)^k} \cdot \left\|x_0 - x^\star\right\|. \tag{A.6}$$

385 In addition, if the step size $s = \theta/\mu$, where $\theta > 0$, we have

$$\left\|x_k - x^\star\right\| \leq \frac{1}{(1+\theta)^k}\left\|x_0 - x^\star\right\|. \tag{A.7}$$

386 *Proof.* The Lyapunov function is

$$\mathcal{E}(k) = \left\|x_k - x^\star\right\|^2.$$

387 Then, we calculate the iterate difference as

$$
\begin{aligned}
\mathcal{E}(k+1) - \mathcal{E}(k) &= \left\|x_{k+1} - x^\star\right\|^2 - \left\|x_k - x^\star\right\|^2 \\
&= \left\langle x_{k+1} - x_k, x_{k+1} + x_k - 2x^\star\right\rangle \\
&= -2s\left\langle \nabla f(x_{k+1}), x_{k+1} - x^\star\right\rangle - s^2\left\|\nabla f(x_{k+1})\right\|^2 \\
&\leq -\left(2\mu s + \mu^2 s^2\right)\mathcal{E}(k+1).
\end{aligned}
$$

388 Hence, the proof is complete. $\qquad\square$

389 Next, we show the implicit version of Theorem A.2 as follows.

390 **Theorem A.8.** Let $f \in \mathcal{F}_L^1(\mathbb{R}^n)$. The iterates $\{x_k\}_{k=0}^{\infty}$ generated by implicit gradient descent (A.5)
391 satisfy

$$f(x_k) - f(x^\star) \leq \frac{(1 + Ls)^2 \left(f(x_0) - f(x^\star)\right) \|x_0 - x^\star\|^2}{(1 + Ls)^2 \|x_0 - x^\star\|^2 + ks \left(f(x_0) - f(x^\star)\right)}. \tag{A.8}$$

392 In addition, if the step size is set to $s = \theta/L$, we have

$$f(x_k) - f(x^\star) \leq \frac{L \|x_0 - x^\star\|^2}{2 + k \cdot \frac{1}{\theta + \frac{1}{\theta} + 2}}. \tag{A.9}$$

393 *Proof.* Note that the distance function $\|x_k - x^\star\|^2$ decreases with the iteration number $k$ as

$$\|x_{k+1} - x^\star\|^2 - \|x_k - x^\star\|^2 = -2s \langle \nabla f(x_{k+1}), x_{k+1} - x^\star \rangle - s^2 \|\nabla f(x_{k+1})\|^2$$
$$\leq -s \left(\frac{2}{L} + s\right) \|\nabla f(x_{k+1})\|^2$$
$$\leq 0.$$

394 With the basic convex inequality for $f \in \mathcal{F}_L^1(\mathbb{R}^n)$, we have

$$f(x_{k+1}) - f(x^\star) \leq \langle \nabla f(x_{k+1}), x_{k+1} - x^\star \rangle \leq \|\nabla f(x_{k+1})\| \cdot \|x_0 - x^\star\|.$$

395 Now, the Lyapunov function is defined as

$$\mathcal{E}(k) = f(x_k) - f(x^\star).$$

396 Then we calculate the difference at the $k$th-iteration as

$$\mathcal{E}(k+1) - \mathcal{E}(k) = \left(f(x_{k+1}) - f(x^\star)\right) - \left(f(x_k) - f(x^\star)\right)$$
$$\geq \langle \nabla f(x_{k+1}), x_{k+1} - x_k \rangle - \frac{L}{2} \|x_{k+1} - x_k\|^2$$
$$\geq -s \left(1 + \frac{Ls}{2}\right) \|\nabla f(x_{k+1})\|^2$$
$$\geq -2Ls \left(1 + \frac{Ls}{2}\right) \mathcal{E}(k+1)$$

397 and

$$\mathcal{E}(k+1) - \mathcal{E}(k) = \left(f(x_{k+1}) - f(x^\star)\right) - \left(f(x_k) - f(x^\star)\right)$$
$$\leq \langle \nabla f(x_{k+1}), x_{k+1} - x_k \rangle$$
$$= -s \cdot \|\nabla f(x_{k+1})\|^2$$
$$\leq -s \cdot \frac{\mathcal{E}(k+1)^2}{\|x_0 - x^\star\|_2^2}$$
$$\leq -s \cdot \frac{\mathcal{E}(k+1)}{\mathcal{E}(k)} \cdot \frac{\mathcal{E}(k)\mathcal{E}(k+1)}{\|x_0 - x^\star\|^2}$$
$$\leq -\frac{s}{(1 + Ls)^2} \cdot \frac{\mathcal{E}(k)\mathcal{E}(k+1)}{\|x_0 - x^\star\|^2}.$$

398 Hence, the convergence rate is given as

$$f(x_k) - f(x^\star) \leq \frac{(1 + Ls)^2 \left(f(x_0) - f(x^\star)\right) \|x_0 - x^\star\|^2}{(1 + Ls)^2 \|x_0 - x^\star\|^2 + ks \left(f(x_0) - f(x^\star)\right)}.$$

399 $\qquad\qquad\qquad\qquad\qquad\qquad\qquad\qquad\qquad\qquad\qquad\qquad\qquad\qquad\qquad\qquad\qquad\qquad\quad\square$

400 Finally, we present the implicit version of Theorem A.3.

**Theorem A.9.** Let $f \in \mathcal{F}_L^1(\mathbb{R}^n)$. The iterates $\{x_k\}_{k=0}^{\infty}$ generated by implicit gradient descent (A.5) satisfy

$$
\begin{cases}
f(x_k) - f(x^\star) \leq \dfrac{\|x_0 - x^\star\|^2}{2ks} \\[2mm]
\min\limits_{0 \leq i \leq k} \|\nabla f(x_i)\|^2 \leq \dfrac{2\|x_0 - x^\star\|^2}{s^2(k+1)(k+2)}.
\end{cases}
\tag{A.10}
$$

In addition, if the step size is set to $s = 1/L$, we have

$$
\begin{cases}
f(x_k) - f(x^\star) \leq \dfrac{L\|x_0 - x^\star\|^2}{2k} \\[2mm]
\min\limits_{0 \leq i \leq k} \|\nabla f(x_i)\|^2 \leq \dfrac{2L^2\|x_0 - x^\star\|^2}{(k+1)(k+2)}.
\end{cases}
\tag{A.11}
$$

*Proof.* The Lyapunov function is

$$
\mathcal{E}(k) = ks\left(f(x_k) - f(x^\star)\right) + \frac{1}{2}\|x_k - x^\star\|^2.
$$

Then, we calculate the iterate difference as

$$
\begin{aligned}
&\mathcal{E}(k+1) - \mathcal{E}(k) \\
&= s\left(f(x_{k+1}) - f(x^\star)\right) + ks\left(f(x_{k+1}) - f(x_k)\right) + \frac{1}{2}\langle x_{k+1} - x_k, x_{k+1} + x_k - 2x^\star \rangle \\
&\leq s\left(f(x_{k+1}) - f(x^\star) - \langle \nabla f(x_{k+1}), x_{k+1} - x^\star \rangle\right) \\
&\quad + ks\langle \nabla f(x_{k+1}), x_{k+1} - x_k \rangle - \frac{1}{2}\|x_{k+1} - x_k\|^2 \\
&\leq -s^2\left(\frac{1}{2Ls} + k + \frac{1}{2}\right)\|\nabla f(x_{k+1})\|^2 \\
&\leq -\frac{s^2}{2}(k+1)\|\nabla f(x_{k+1})\|^2.
\end{aligned}
$$

Hence, the proof is complete. $\qquad\square$

# B  Proofs for Section 3

Here, we first describe and prove Theorem B.1 below. Then we complete the proof of Theorem 3.1 by viewing it as a special case of Theorem B.1.

**Theorem B.1** (Discretization of NAG-SC ODE — General). For any $f \in \mathcal{S}_{\mu,L}^1(\mathbb{R}^n)$, the following conclusions hold:

(a) Taking $0 < s \leq 4/(9L)$, the symplectic Euler scheme satisfies

$$
\begin{aligned}
&f(x_k) - f(x^\star) \\
&\leq \left(\frac{sL\left(2 + (1 + 3\sqrt{\mu s})^2\right)}{(1+\sqrt{\mu s})^2} + \frac{2\mu}{L} + \frac{1+\sqrt{\mu s}}{2} - \frac{sL(1+\sqrt{\mu s})^2}{2(1+2\sqrt{\mu s})}\right) \frac{L\|x_0 - x^\star\|^2}{\left(1 + \frac{\sqrt{\mu s}}{6}\right)^k}.
\end{aligned}
\tag{B.1}
$$

(b) Taking $0 < s \leq \mu/(100L^2)$, the explicit Euler scheme satisfies

$$
\begin{aligned}
&f(x_k) - f(x^\star) \\
&\leq \left(\frac{3 - 2\sqrt{\mu s} + \mu s}{2 + 4\sqrt{\mu s} + 2\mu s} \cdot sL + \frac{2\mu}{L} + \frac{1+\sqrt{\mu s}}{2}\right) L\|x_0 - x^\star\|^2 \left(1 - \frac{\sqrt{\mu s}}{8}\right)^k.
\end{aligned}
\tag{B.2}
$$

(c) Taking $0 < s \leq 1/L$, the implicit Euler scheme satisfies

$$
f(x_k) - f(x^\star) \leq \left(\frac{3 - 2\sqrt{\mu s} + \mu s}{2 + 4\sqrt{\mu s} + 2\mu s} \cdot sL + \frac{2\mu}{L} + \frac{1+\sqrt{\mu s}}{2}\right) \frac{L\|x_0 - x^\star\|^2}{\left(1 + \frac{\sqrt{\mu s}}{4}\right)^k}.
\tag{B.3}
$$

## B.1 Proof of Theorem B.1

(a) The Lyapunov function is constructed as

$$\mathcal{E}(k) = \frac{1}{4} \left\| v_k \right\|^2 + \frac{1}{4} \left\| 2\sqrt{\mu}(x_{k+1} - x^\star) + v_k + \sqrt{s}\nabla f(x_k) \right\|^2$$
$$+ (1 + \sqrt{\mu s}) \left( f(x_k) - f(x^\star) \right) - \frac{(1 + \sqrt{\mu s})^2}{1 + 2\sqrt{\mu s}} \cdot \frac{s}{2} \left\| \nabla f(x_k) \right\|^2.$$

With the basic inequality for $f \in \mathcal{S}_{\mu,L}^1(\mathbb{R}^n)$

$$f(x_{k+1}) - f(x_k) \leq \langle \nabla f(x_{k+1}), x_{k+1} - x_k \rangle - \frac{1}{2L} \left\| \nabla f(x_{k+1}) - \nabla f(x_k) \right\|^2,$$

then the iterate difference can be calculated as

$$\mathcal{E}(k+1) - \mathcal{E}(k)$$
$$= \frac{1}{4} \langle v_{k+1} - v_k, v_{k+1} + v_k \rangle + (1 + \sqrt{\mu s}) \left( f(x_{k+1}) - f(x_k) \right)$$
$$+ \frac{1}{4} \langle 2\sqrt{\mu}(x_{k+2} - x_{k+1}) + v_{k+1} - v_k + \sqrt{s} \left( \nabla f(x_{k+1}) - \nabla f(x_k) \right),$$
$$2\sqrt{\mu} \left( x_{k+2} + x_{k+1} - 2x^\star \right) + v_{k+1} + v_k$$
$$+ \sqrt{s} \left( \nabla f(x_{k+1}) + \nabla f(x_k) \right) \rangle$$
$$- \frac{(1 + \sqrt{\mu s})^2}{1 + 2\sqrt{\mu s}} \cdot \frac{s}{2} \left( \left\| \nabla f(x_{k+1}) \right\|^2 - \left\| \nabla f(x_k) \right\|^2 \right)$$
$$\leq -\sqrt{\mu s} \left\| v_{k+1} \right\|^2 - \frac{\sqrt{s}}{2(1 + \sqrt{\mu s})} \langle \nabla f(x_{k+1}) - \nabla f(x_k), v_k \rangle$$
$$+ \frac{s}{2 \left( 1 + 2\sqrt{\mu s} \right)} \left\| \nabla f(x_{k+1}) - \nabla f(x_k) \right\|^2$$
$$+ \frac{s}{2} \cdot \frac{1 + \sqrt{\mu s}}{1 + 2\sqrt{\mu s}} \langle \nabla f(x_{k+1}) - \nabla f(x_k), \nabla f(x_{k+1}) \rangle$$
$$- \frac{\sqrt{s} \left( 1 + \sqrt{\mu s} \right)}{2} \langle \nabla f(x_{k+1}), v_{k+1} \rangle - \frac{1}{4} \left\| v_{k+1} - v_k \right\|^2$$
$$+ (1 + \sqrt{\mu s}) \sqrt{s} \langle \nabla f(x_{k+1}), v_k \rangle - \frac{1 + \sqrt{\mu s}}{2L} \left\| \nabla f(x_{k+1}) - \nabla f(x_k) \right\|^2$$
$$- \frac{1}{2} \langle (1 + \sqrt{\mu s})\sqrt{s}\nabla f(x_{k+1}),$$
$$(1 + 2\sqrt{\mu s}) \, v_{k+1} + 2\sqrt{\mu}(x_{k+1} - x^\star) + \sqrt{s}\nabla f(x_{k+1}) \rangle$$
$$- \frac{1}{4} (1 + \sqrt{\mu s})^2 \, s \left\| \nabla f(x_{k+1}) \right\|^2 - \frac{s}{2} \left( \left\| \nabla f(x_{k+1}) \right\|^2 - \left\| \nabla f(x_k) \right\|^2 \right)$$
$$\leq -\sqrt{\mu s} \left( \left\| v_{k+1} \right\|^2 + (1 + \sqrt{\mu s}) \langle \nabla f(x_{k+1}), x_{k+1} - x^\star \rangle \right)$$
$$- \left( \frac{1 + \sqrt{\mu s}}{2} \right) \left[ \sqrt{s} \langle \nabla f(x_{k+1}), (1 + 2\sqrt{\mu s}) \, v_{k+1} - v_k \rangle + s \left\| \nabla f(x_{k+1}) \right\|^2 \right]$$
$$- \frac{\sqrt{s}}{2(1 + \sqrt{\mu s})} \langle \nabla f(x_{k+1}) - \nabla f(x_k), v_k \rangle$$
$$+ \frac{s}{2} \cdot \frac{1 + \sqrt{\mu s}}{1 + 2\sqrt{\mu s}} \langle \nabla f(x_{k+1}) - \nabla f(x_k), \nabla f(x_{k+1}) \rangle$$
$$- \frac{1}{4} \left[ \left\| v_{k+1} - v_k \right\|^2 + (1 + \sqrt{\mu s})^2 \, s \left\| \nabla f(x_{k+1}) \right\|^2 \right.$$
$$\left. + 2(1 + \sqrt{\mu s})\sqrt{s} \langle \nabla f(x_{k+1}), v_{k+1} - v_k \rangle \right]$$
$$- \frac{1}{2} \left( \frac{1 + \sqrt{\mu s}}{L} - \frac{s}{1 + 2\sqrt{\mu s}} \right) \left\| \nabla f(x_{k+1}) - \nabla f(x_k) \right\|^2$$

$$-\frac{(1+\sqrt{\mu s})^2}{1+2\sqrt{\mu s}} \cdot \frac{s}{2} \left( \|\nabla f(x_{k+1})\|^2 - \|\nabla f(x_k)\|^2 \right).$$

Noting the following two inequalities

$$-\frac{1}{4} \Big[ \|v_{k+1} - v_k\|^2 + (1+\sqrt{\mu s})^2 \, s \, \|\nabla f(x_{k+1})\|^2$$
$$+ 2(1+\sqrt{\mu s})\sqrt{s} \, \langle \nabla f(x_{k+1}), v_{k+1} - v_k \rangle \Big]$$
$$= -\frac{1}{4} \left\| v_{k+1} - v_k + (1+\sqrt{\mu s}) \, \sqrt{s} \nabla f(x_k) \right\|^2 \leq 0,$$

and

$$-\frac{1}{2} (1+\sqrt{\mu s}) \left[ \sqrt{s} \, \langle \nabla f(x_{k+1}), (1+2\sqrt{\mu s}) \, v_{k+1} - v_k \rangle + s \, \|\nabla f(x_{k+1})\|^2 \right]$$
$$= -\left( \frac{1+\sqrt{\mu s}}{2} \right) [\sqrt{s}\langle \nabla f(x_{k+1}),$$
$$-\sqrt{s} \left( \nabla f(x_{k+1}) - \nabla f(x_k) \right) - \sqrt{s}(1+\sqrt{\mu s})\nabla f(x_{k+1}) \rangle$$
$$+ s \, \|\nabla f(x_{k+1})\|^2]$$
$$= \frac{\left(1+\sqrt{\mu s}\right) s}{2} \left( \langle \nabla f(x_{k+1}) - \nabla f(x_k), \nabla f(x_{k+1}) \rangle + \sqrt{\mu s} \, \|\nabla f(x_{k+1})\|^2 \right),$$

we see that the iterate difference is

$$\mathcal{E}(k+1) - \mathcal{E}(k)$$
$$\leq -\sqrt{\mu s} \left[ \|v_{k+1}\|^2 + (1+\sqrt{\mu s}) \left( \langle \nabla f(x_{k+1}), x_{k+1} - x^\star \rangle - \frac{s}{2} \, \|\nabla f(x_{k+1})\|^2 \right) \right]$$
$$- \frac{1}{2(1+\sqrt{\mu s})} \, \langle \nabla f(x_{k+1}) - \nabla f(x_k), x_{k+1} - x_k \rangle$$
$$+ \frac{(1+\sqrt{\mu s})^2}{1+2\sqrt{\mu s}} \cdot s \, \langle \nabla f(x_{k+1}), \nabla f(x_{k+1}) - \nabla f(x_k) \rangle$$
$$- \frac{1}{2} \left( \frac{1+\sqrt{\mu s}}{L} - \frac{s}{1+2\sqrt{\mu s}} \right) \|\nabla f(x_{k+1}) - \nabla f(x_k)\|^2$$
$$- \frac{(1+\sqrt{\mu s})^2}{1+2\sqrt{\mu s}} \cdot \frac{s}{2} \left( \|\nabla f(x_{k+1})\|^2 - \|\nabla f(x_k)\|^2 \right)$$
$$\leq -\sqrt{\mu s} \left[ \|v_{k+1}\|^2 + (1+\sqrt{\mu s}) \left( \langle \nabla f(x_{k+1}), x_{k+1} - x^\star \rangle - \frac{s}{2} \, \|\nabla f(x_{k+1})\|^2 \right) \right]$$
$$- \frac{1}{2(1+\sqrt{\mu s})} \, \langle \nabla f(x_{k+1}) - \nabla f(x_k), x_{k+1} - x_k \rangle$$
$$+ \frac{1}{2} \left( \frac{1}{1+2\sqrt{\mu s}} + \frac{(1+\sqrt{\mu s})^2}{1+2\sqrt{\mu s}} - \frac{1+\sqrt{\mu s}}{Ls} \right) s \, \|\nabla f(x_{k+1}) - \nabla f(x_k)\|^2 .$$

Furthermore, taking the basic inequality for $f \in \mathcal{S}_{\mu,L}^1(\mathbb{R}^n)$

$$\langle \nabla f(x_{k+1}) - \nabla f(x_k), x_{k+1} - x_k \rangle \geq \frac{1}{L} \, \|\nabla f(x_{k+1}) - \nabla f(x_k)\|^2 ,$$

the iterate difference can be calculated as

$$\mathcal{E}(k+1) - \mathcal{E}(k)$$
$$\leq -\sqrt{\mu s} \left[ \|v_{k+1}\|^2 + (1+\sqrt{\mu s}) \left( \langle \nabla f(x_{k+1}), x_{k+1} - x^\star \rangle - \frac{s}{2} \, \|\nabla f(x_{k+1})\|^2 \right) \right]$$
$$- \frac{2 + 2\sqrt{\mu s} + \mu s}{2 \left(1 + 2\sqrt{\mu s}\right)} \left( \frac{1}{L} - s \right) \|\nabla f(x_{k+1}) - \nabla f(x_k)\|^2 .$$

Next, we consider how to set the step size $s$. First, when the step size satisfies $s \leq 1/L$, we have

$$\mathcal{E}(k+1) - \mathcal{E}(k)$$

$$\leq -\sqrt{\mu s}\left[\|v_{k+1}\|^2 + (1+\sqrt{\mu s})\left(\langle \nabla f(x_{k+1}), x_{k+1} - x^\star\rangle - \frac{s}{2}\|\nabla f(x_{k+1})\|^2\right)\right].$$

Noting the basic inequality for $f \in \mathcal{S}_{\mu,L}^1(\mathbb{R}^n)$

$$f(x_{k+1}) - f(x^\star) \leq \langle \nabla f(x_{k+1}), x_{k+1} - x^\star\rangle - \frac{1}{2L}\|\nabla f(x_{k+1})\|^2,$$

the iterate difference can be obtained as

$$\mathcal{E}(k+1) - \mathcal{E}(k)$$
$$\leq -\sqrt{\mu s}\left[(f(x_{k+1}) - f(x^\star)) + \|v_{k+1}\|^2 + \frac{\mu}{2}\|x_k - x^\star\|^2\right.$$
$$\left. + \sqrt{\mu s}\left(f(x_{k+1}) - f(x^\star) - \frac{s}{2}\|\nabla f(x_{k+1})\|^2\right)\right].$$

Furthermore, using the Cauchy-Schwarz inequality

$$\left\|2\sqrt{\mu}\,(x_{k+1} - x^\star) + v_k + \sqrt{s}\nabla f(x_k)\right\|^2$$
$$= \left\|2\sqrt{\mu}(x_k - x^\star) + (1 + 2\sqrt{\mu s})v_k + \sqrt{s}\nabla f(x_k)\right\|^2$$
$$\leq 3\left(4\mu\|x_k - x^\star\|^2 + (1 + 2\sqrt{\mu s})^2\|v_k\|^2 + s\|\nabla f(x_k)\|^2\right),$$

and the following basic inequality for $f \in \mathcal{S}_{\mu,L}^1(\mathbb{R}^n)$

$$\frac{3s}{4}\|\nabla f(x_k)\|^2 - \frac{(1+\sqrt{\mu s})^2}{1 + 2\sqrt{\mu s}}\cdot\frac{s}{2}\|\nabla f(x_k)\|^2$$
$$\leq \frac{Ls}{2}\left(f(x_k) - f(x^\star)\right) - \frac{\mu s^2}{2\left(1 + 2\sqrt{\mu s}\right)}\|\nabla f(x_k)\|^2,$$

the Lyapunov function satisfies

$$\mathcal{E}(k) \leq \left(1 + \sqrt{\mu s} + \frac{Ls}{2}\right)(f(x_k) - f(x^\star)) + (1 + 3\sqrt{\mu s} + 3\mu s)\|v_k\|^2$$
$$+ 3\mu\|x_k - x^\star\|^2 + \frac{\mu s}{1 + 2\sqrt{\mu s}}\left(f(x_k) - f(x^\star) - \frac{s}{2}\|\nabla f(x_k)\|^2\right).$$

Therefore, when $s \leq 4/(9L)$, the iterate difference for the Lyapunov function satisfies

$$\mathcal{E}(k+1) - \mathcal{E}(k) \leq -\frac{\sqrt{\mu s}}{6}\mathcal{E}(k+1).$$

Hence, the proof is complete.

(b) The Lyapunov function is

$$\mathcal{E}(k) = \frac{1}{4}\|v_k\|^2 + (1 + \sqrt{\mu s})(f(x_k) - f(x^\star))$$
$$+ \frac{1}{4}\left\|2\sqrt{\mu}(x_k - x^\star) + v_k + \sqrt{s}\nabla f(x_k)\right\|^2.$$

With the basic inequality for $f \in \mathcal{S}_{\mu,L}^1(\mathbb{R}^n)$

$$f(x_{k+1}) - f(x_k) \leq \langle \nabla f(x_k), x_{k+1} - x_k\rangle + \frac{L}{2}\|x_{k+1} - x_k\|^2,$$

we can calculate the iterate difference

$$\mathcal{E}(k+1) - \mathcal{E}(k)$$
$$= \frac{1}{4}\langle v_{k+1} - v_k, v_{k+1} + v_k\rangle + (1 + \sqrt{\mu s})(f(x_{k+1}) - f(x_k))$$
$$+ \frac{1}{4}\langle 2\sqrt{\mu}(x_{k+1} - x_k) + v_{k+1} - v_k + \sqrt{s}\left(\nabla f(x_{k+1}) - \nabla f(x_k)\right),$$
$$2\sqrt{\mu}\,(x_{k+1} + x_k - 2x^\star) + v_{k+1} + v_k + \sqrt{s}(\nabla f(x_{k+1}) + \nabla f(x_k)\rangle$$

$$\leq \frac{1}{2} \langle v_{k+1} - v_k, v_k \rangle + \frac{1}{4} \|v_{k+1} - v_k\|^2$$

$$+ (1 + \sqrt{\mu s}) \left( \langle \nabla f(x_k), x_{k+1} - x_k \rangle + \frac{L}{2} \|x_{k+1} - x_k\|^2 \right)$$

$$- \frac{1}{2} \langle (1 + \sqrt{\mu s}) \sqrt{s} \nabla f(x_k), 2\sqrt{\mu}(x_k - x^\star) + v_k + \sqrt{s} \nabla f(x_k) \rangle$$

$$+ \frac{1}{4} \left\| (1 + \sqrt{\mu s}) \sqrt{s} \nabla f(x_k) \right\|^2$$

$$= -\sqrt{\mu s} \|v_k\|^2 - \frac{1}{2} \langle \nabla f(x_{k+1}) - \nabla f(x_k), x_{k+1} - x_k \rangle$$

$$+ \frac{1}{4} \left\| 2\sqrt{\mu s} v_k + \sqrt{s} \left( \nabla f(x_{k+1}) - \nabla f(x_k) \right) + \sqrt{s} \left( 1 + \sqrt{\mu s} \right) \nabla f(x_k) \right\|^2$$

$$+ \frac{(1 + \sqrt{\mu s}) s L}{2} \|v_k\|^2 - \sqrt{\mu s} \left( 1 + \sqrt{\mu s} \right) \langle \nabla f(x_k), x_k - x^\star \rangle$$

$$- \frac{\left( 1 + \sqrt{\mu s} \right) s}{2} \|\nabla f(x_k)\|^2 + \frac{1}{4} \left( 1 + \sqrt{\mu s} \right)^2 s \|\nabla f(x_k)\|^2 .$$

436      Using the Cauchy-Schwartz inequality

$$\left\| 2\sqrt{\mu s} v_k + \sqrt{s} \left( \nabla f(x_{k+1}) - \nabla f(x_k) \right) + \sqrt{s} \left( 1 + \sqrt{\mu s} \right) \nabla f(x_k) \right\|^2$$

$$\leq 12 \mu s \|v_k\|^2 + 3s \|\nabla f(x_{k+1}) - \nabla f(x_k)\|^2 + 3s \left( 1 + \sqrt{\mu s} \right)^2 \|\nabla f(x_0)\|^2 ,$$

437      the iterate difference for the Lyapunov function can be calculated as

$$\mathcal{E}(k+1) - \mathcal{E}(k)$$

$$\leq -\sqrt{\mu s} \left( \|v_k\|^2 + (1 + \sqrt{\mu s}) \langle \nabla f(x_k), x_k - x^\star \rangle + \frac{s}{2} \|\nabla f(x_k)\|^2 \right)$$

$$- \frac{1}{2} \langle \nabla f(x_{k+1}) - \nabla f(x_k), x_{k+1} - x_k \rangle + \frac{3s}{4} \|\nabla f(x_{k+1}) - \nabla f(x_k)\|^2$$

$$+ \left( 3\mu s + \frac{(1 + \sqrt{\mu s}) s L}{2} \right) \|v_k\|^2 + \left[ (1 + \sqrt{\mu s})^2 - \frac{1}{2} \right] s \|\nabla f(x_k)\|^2 .$$

438      Furthermore, combined with the basic inequality for $f \in \mathcal{S}_{\mu,L}^1(\mathbb{R}^n)$,

$$\begin{cases} \|\nabla f(x_{k+1}) - \nabla f(x_k)\|^2 \leq L \langle \nabla f(x_{k+1}) - \nabla f(x_k), x_{k+1} - x_k \rangle \\ \|\nabla f(x_k)\|^2 \leq L \langle \nabla f(x_k), x_k - x^\star \rangle , \end{cases}$$

439      the iterate difference for the Lyapunov function can be calculated as

$$\mathcal{E}(k+1) - \mathcal{E}(k)$$

$$\leq -\frac{\sqrt{\mu s}}{2} \left( \|v_k\|^2 + (1 + \sqrt{\mu s}) \langle \nabla f(x_k), x_k - x^\star \rangle + \frac{s}{2} \|\nabla f(x_k)\|^2 \right)$$

$$- \left( \frac{1}{2L} - \frac{3s}{4} \right) \|\nabla f(x_{k+1}) - \nabla f(x_k)\|^2$$

$$- \left( \frac{\sqrt{\mu s}}{2} - 3\mu s - \frac{(1 + \sqrt{\mu s}) s L}{2} \right) \|v_k\|^2$$

$$- \left[ \frac{\sqrt{\mu s} \left( 1 + \sqrt{\mu s} \right)}{2L} - \left( \frac{1}{2} + \sqrt{\mu s} \right) \left( 1 + \sqrt{\mu s} \right) s \right] \|\nabla f(x_k)\|^2 .$$

440      Simple calculation tells us when the step size satisfies $s \leq \mu/(100L^2)$, we have

$$\mathcal{E}(k+1) - \mathcal{E}(k)$$

$$\leq -\frac{\sqrt{\mu s}}{2} \left( \|v_k\|^2 + (1 + \sqrt{\mu s}) \langle \nabla f(x_k), x_k - x^\star \rangle + \frac{s}{2} \|\nabla f(x_k)\|^2 \right) .$$

441      Furthermore, taking the Cauchy-Schwartz inequality

$$\left\|2\sqrt{\mu}\left(x_{k+1}-x^{\star}\right)+v_k+\sqrt{s}\nabla f(x_k)\right\|^2$$
$$\leq 3\left(4\mu\left\|x_k-x^{\star}\right\|^2+\left\|v_k\right\|^2+s\left\|\nabla f(x_k)\right\|^2\right),$$

we can obtain the final estimate for the iterate difference

$$\mathcal{E}(k+1)-\mathcal{E}(k)\leq-\frac{\sqrt{\mu s}}{8}\mathcal{E}(k).$$

Hence, the proof is complete.

(c) The Lyapunov function is

$$\mathcal{E}(k)=\frac{1}{4}\left\|v_k\right\|^2+\left(1+\sqrt{\mu s}\right)\left(f(x_k)-f(x^{\star})\right)$$
$$+\frac{1}{4}\left\|2\sqrt{\mu}(x_k-x^{\star})+v_k+\sqrt{s}\nabla f(x_k)\right\|^2$$

With the Cauchy-Schwartz inequality

$$\left\|2\sqrt{\mu}\left(x_k-x^{\star}\right)+v_k+\sqrt{s}\nabla f(x_k)\right\|^2$$
$$\leq 3\left(4\mu\left\|x_k-x^{\star}\right\|^2+\left\|v_k\right\|^2+s\left\|\nabla f(x_k)\right\|^2\right),$$

and the basic inequality for $f\in\mathcal{S}_{\mu,L}^1(\mathbb{R}^n)$

$$\begin{cases}f(x_{k+1})-f(x_k)\leq\langle\nabla f(x_{k+1}),x_{k+1}-x_k\rangle\\f(x^{\star})\geq f(x_{k+1})+\langle\nabla f(x_{k+1}),x^{\star}-x_{k+1}\rangle+\dfrac{\mu}{2}\left\|x_{k+1}-x^{\star}\right\|^2\\\langle\nabla f(x_{k+1})-\nabla f(x_k),x_{k+1}-x_k\rangle\geq\mu\left\|x_{k+1}-x_k\right\|^2\geq 0,\end{cases}$$

we can calculate the iterate difference

$$\mathcal{E}(k+1)-\mathcal{E}(k)$$
$$=\frac{1}{4}\langle v_{k+1}-v_k,v_{k+1}+v_k\rangle+\left(1+\sqrt{\mu s}\right)\left(f(x_{k+1})-f(x_k)\right)$$
$$+\frac{1}{4}\langle 2\sqrt{\mu}(x_{k+1}-x_k)+v_{k+1}-v_k+\sqrt{s}\left(\nabla f(x_{k+1})-\nabla f(x_k)\right)$$
$$2\sqrt{\mu}\left(x_{k+1}+x_k-2x^{\star}\right)+v_{k+1}+v_k+\sqrt{s}\left(\nabla f(x_{k+1})+\nabla f(x_k)\right)\rangle$$
$$\leq\frac{1}{2}\langle v_{k+1}-v_k,v_{k+1}\rangle+\left(1+\sqrt{\mu s}\right)\langle\nabla f(x_{k+1}),x_{k+1}-x_k\rangle$$
$$-\frac{1}{2}\langle(1+\sqrt{\mu s})\sqrt{s}\nabla f(x_{k+1}),2\sqrt{\mu}(x_{k+1}-x^{\star})+v_{k+1}+\sqrt{s}\nabla f(x_{k+1})\rangle$$
$$-\frac{1}{4}\left\|v_{k+1}-v_k\right\|^2-\frac{1}{4}\left\|(1+\sqrt{\mu s})\sqrt{s}\nabla f(x_{k+1})\right\|^2$$
$$\leq-\sqrt{\mu s}\left(\left\|v_{k+1}\right\|^2+\left(1+\sqrt{\mu s}\right)\langle\nabla f(x_{k+1}),x_{k+1}-x^{\star}\rangle+\frac{s}{2}\left\|\nabla f(x_{k+1})\right\|^2\right)$$
$$-\frac{1}{2}\langle\nabla f(x_{k+1})-\nabla f(x_k),x_{k+1}-x_k\rangle-\frac{s}{2}\left\|\nabla f(x_{k+1})\right\|^2$$
$$\leq-\frac{\sqrt{\mu s}}{4}\mathcal{E}(k+1).$$

Hence, the proof is complete.

## B.2 Proof of Theorem 3.2

Here, we first describe and prove Theorem B.2 below. Then we complete the proof of Theorem 3.2 by viewing it as a special case of Theorem B.2.

**Theorem B.2** (Discretization of heavy-ball ODE — General). For any $f\in\mathcal{S}_{\mu,L}^1(\mathbb{R}^n)$, the following conclusions hold:

(a) Taking $0 < s \leq \mu/(16L^2)$, the symplectic Euler scheme satisfies

$$f(x_k) - f(x^\star) \leq \left( \frac{(3 + 8\sqrt{\mu s} + 8\mu s)\, sL}{(1 + \sqrt{\mu s})^2} + \frac{2\mu}{L} + \frac{1 + \sqrt{\mu s}}{2} \right) \frac{L \|x_0 - x^\star\|^2}{\left(1 + \frac{\sqrt{\mu s}}{4}\right)^k}. \quad \text{(B.4)}$$

(b) Taking $0 < s \leq \mu/(36L^2)$, the explicit Euler scheme satisfies

$$f(x_k) - f(x^\star)$$
$$\leq \left( \frac{3sL}{(1 + \sqrt{\mu s})^2} + \frac{2\mu}{L} + \frac{1 + \sqrt{\mu s}}{2} \right) L \|x_0 - x^\star\|^2 \left(1 - \frac{\sqrt{\mu s}}{8}\right)^k. \quad \text{(B.5)}$$

(c) Taking $0 < s \leq 1/L$, the implicit Euler scheme satisfies

$$f(x_k) - f(x^\star) \leq \left( \frac{3sL}{(1 + \sqrt{\mu s})^2} + \frac{2\mu}{L} + \frac{1 + \sqrt{\mu s}}{2} \right) \frac{L \|x_0 - x^\star\|^2}{\left(1 + \frac{\sqrt{\mu s}}{4}\right)^k}. \quad \text{(B.6)}$$

## Proof of Theorem B.2

(a) The Lyapunov function is

$$\mathcal{E}(k) = \frac{1}{4} \|v_k\|^2 + \frac{1}{4} \|2\sqrt{\mu}(x_k - x^\star) + v_k\|^2 + (1 + \sqrt{\mu s})\, (f(x_k) - f(x^\star)).$$

With the Cauchy-Schwartz inequality

$$\|2\sqrt{\mu}(x_k - x^\star) + v_k\|^2 \leq 2 \left( 4\mu \|x_k - x^\star\|^2 + \|v_k\|^2 \right),$$

and the basic inequality for $f \in \mathcal{S}^1_{\mu,L}(\mathbb{R}^n)$

$$\begin{cases} f(x_{k+1}) - f(x_k) \leq \langle \nabla f(x_{k+1}), x_{k+1} - x_k \rangle \\ f(x^\star) \geq f(x_{k+1}) + \langle \nabla f(x_{k+1}), x^\star - x_{k+1} \rangle + \frac{\mu}{2} \|x_{k+1} - x^\star\|^2, \end{cases}$$

then we can calculate the iterative difference

$$\mathcal{E}(k+1) - \mathcal{E}(k)$$
$$= \frac{1}{4} \langle v_{k+1} - v_k, v_{k+1} + v_k \rangle + (1 + \sqrt{\mu s})\, (f(x_{k+1}) - f(x_k))$$
$$\quad + \frac{1}{4} \langle 2\sqrt{\mu}(x_{k+1} - x_k) + v_{k+1} - v_k, 2\sqrt{\mu}(x_{k+1} + x_k - 2x^\star) + v_{k+1} + v_k \rangle$$
$$\leq \frac{1}{2} \langle v_{k+1} - v_k, v_{k+1} \rangle + (1 + \sqrt{\mu s})\, \langle \nabla f(x_{k+1}), x_{k+1} - x_k \rangle$$
$$\quad + \frac{1}{2} \langle 2\sqrt{\mu}(x_{k+1} - x_k) + v_{k+1} - v_k, 2\sqrt{\mu}(x_{k+1} - x^\star) + v_{k+1} \rangle$$
$$\quad - \frac{1}{4} \|v_{k+1} - v_k\|^2 - \frac{1}{4} \|2\sqrt{\mu}(x_{k+1} - x_k) + v_{k+1} - v_k\|^2$$
$$\leq -\sqrt{\mu s} \left[ \|v_{k+1}\|^2 + (1 + \sqrt{\mu s})\, \langle \nabla f(x_{k+1}), x_{k+1} - x^\star \rangle \right]$$
$$\leq -\sqrt{\mu s} \left[ \|v_{k+1}\|^2 + (1 + \sqrt{\mu s})\, (f(x) - f(x^\star)) + \frac{\mu}{2} \|x_{k+1} - x^\star\|^2 \right]$$
$$\leq -\frac{\sqrt{\mu s}}{4} \mathcal{E}(k+1).$$

Hence, the proof is complete.

(b) The Lyapunov function is

$$\mathcal{E}(k) = \frac{1}{4} \|v_k\|^2 + \frac{1}{4} \|2\sqrt{\mu}(x_k - x^\star) + v_k\|^2 + (1 + \sqrt{\mu s})\, (f(x_k) - f(x^\star)).$$

With the Cauchy-Schwartz inequality

$$\|2\sqrt{\mu}(x_k - x^\star) + v_k\|^2 \leq 2\left(4\mu \|x_k - x^\star\|^2 + \|v_k\|^2\right),$$

and the basic inequality for $f \in \mathcal{S}_{\mu,L}^1(\mathbb{R}^n)$

$$\begin{cases} f(x_{k+1}) - f(x_k) \leq \langle \nabla f(x_k), x_{k+1} - x_k \rangle + \dfrac{L}{2} \|x_{k+1} - x_k\|^2 \\ f(x^\star) \geq f(x_{k+1}) + \langle \nabla f(x_{k+1}), x^\star - x_{k+1} \rangle + \dfrac{\mu}{2} \|x^\star - x_{k+1}\|^2, \end{cases}$$

then we calculate the iterative difference

$$\begin{aligned}
\mathcal{E}&(k+1) - \mathcal{E}(k) \\
&= \frac{1}{4} \langle v_{k+1} - v_k, v_{k+1} + v_k \rangle + (1 + \sqrt{\mu s})\left(f(x_{k+1}) - f(x_k)\right) \\
&\quad + \frac{1}{4} \langle 2\sqrt{\mu}(x_{k+1} - x_k) + v_{k+1} - v_k, 2\sqrt{\mu}(x_{k+1} + x_k - 2x^\star) + v_{k+1} + v_k \rangle \\
&\leq \frac{1}{2} \langle v_{k+1} - v_k, v_k \rangle + (1 + \sqrt{\mu s}) \langle \nabla f(x_k), x_{k+1} - x_k \rangle \\
&\quad + \frac{\left(1 + \sqrt{\mu s}\right) L}{2} \|x_{k+1} - x_k\|^2 \\
&\quad + \frac{1}{2} \langle 2\sqrt{\mu}(x_{k+1} - x_k) + v_{k+1} - v_k, 2\sqrt{\mu}(x_k - x^\star) + v_k \rangle \\
&\quad + \frac{1}{4} \|v_{k+1} - v_k\|^2 + \frac{1}{4} \|2\sqrt{\mu}(x_{k+1} - x_k) + v_{k+1} - v_k\|^2 \\
&\leq -\sqrt{\mu s}\left(\|v_k\|^2 + (1 + \sqrt{\mu s}) \langle \nabla f(x_k), x_k - x^\star \rangle \right) \\
&\quad + \frac{\left(1 + \sqrt{\mu s}\right) L s}{2} \|v_k\|^2 + \frac{s}{4} \|2\sqrt{\mu} v_k + \nabla f(x_k)\|^2 + \frac{s}{4} \|\nabla f(x_k)\|^2 \\
&\leq -\frac{\sqrt{\mu s}}{2}\left(\|v_k\|^2 + f(x_k) - f(x^\star) + \frac{\mu}{2} \|x_k - x^\star\|^2\right) \\
&\quad - \frac{\sqrt{\mu s}}{2}\left(\|v_k\|^2 + \frac{\left(1 + \sqrt{\mu s}\right)}{L} \|\nabla f(x_k)\|^2\right) \\
&\quad + s\left(2\mu + \frac{L\left(1 + \sqrt{\mu s}\right)}{2}\right) \|v_k\|^2 + \frac{3s}{4} \|\nabla f(x_k)\|^2.
\end{aligned}$$

Since $\mu \leq L$, then the step size $s \leq \mu/(36L^2)$ satisfies it. Hence, the proof is complete.

(c) The Lyapunov function is constructed as

$$\mathcal{E}(k) = \frac{1}{4} \|v_k\|^2 + \frac{1}{4} \|2\sqrt{\mu}(x_{k+1} - x^\star) + v_k\|^2 + (1 + \sqrt{\mu s})\left(f(x_k) - f(x^\star)\right).$$

With Cauchy-Schwartz inequality

$$\begin{aligned}
\|2\sqrt{\mu}(x_{k+1} - x^\star) + v_k\|^2 &= \|2\sqrt{\mu}(x_k - x^\star) + (1 + 2\sqrt{\mu s})v_k\|^2 \\
&\leq 2\left(4\mu \|x_k - x^\star\|^2 + (1 + 2\sqrt{\mu s})^2 \|v_k\|^2\right),
\end{aligned}$$

and the basic inequality for $f \in \mathcal{S}_{\mu,L}^1(\mathbb{R}^n)$

$$\begin{cases} f(x_{k+1}) - f(x_k) \leq \langle \nabla f(x_{k+1}), x_{k+1} - x_k \rangle - \dfrac{1}{2L} \|\nabla f(x_{k+1}) - \nabla f(x_k)\|^2 \\ f(x^\star) \geq f(x_{k+1}) + \langle \nabla f(x_{k+1}), x^\star - x_{k+1} \rangle + \dfrac{\mu}{2} \|x^\star - x_{k+1}\|^2, \end{cases}$$

then we calculate the iterative difference

$$\mathcal{E}(k+1) - \mathcal{E}(k)$$

$$= \frac{1}{4} \langle v_{k+1} - v_k, v_{k+1} + v_k \rangle + (1 + \sqrt{\mu s}) (f(x_{k+1}) - f(x_k))$$

$$+ \frac{1}{4} \langle 2\sqrt{\mu}(x_{k+2} - x_{k+1}) + v_{k+1} - v_k, 2\sqrt{\mu}(x_{k+2} + x_{k+1} - 2x^\star) + v_{k+1} + v_k \rangle$$

$$\le \frac{1}{2} \langle v_{k+1} - v_k, v_{k+1} \rangle - \frac{1}{4} \|v_{k+1} - v_k\|^2$$

$$+ (1 + \sqrt{\mu s}) \langle \nabla f(x_{k+1}), x_{k+1} - x_k \rangle - \frac{(1 + \sqrt{\mu s})}{2L} \|\nabla f(x_{k+1}) - \nabla f(x_k)\|^2$$

$$+ \frac{1}{2} \langle -\sqrt{s} (1 + \sqrt{\mu s}) \nabla f(x_{k+1}), 2\sqrt{\mu}(x_{k+2} - x^\star) + v_{k+1} \rangle$$

$$- \frac{1}{4} \left\| \sqrt{s} (1 + \sqrt{\mu s}) \nabla f(x_{k+1}) \right\|^2$$

$$\le -\sqrt{\mu s} \left( \|v_{k+1}\|^2 + (1 + \sqrt{\mu s}) \langle \nabla f(x_{k+1}), x_{k+1} - x^\star \rangle \right)$$

$$- \frac{\sqrt{s} (1 + \sqrt{\mu s})}{2} \langle \nabla f(x_{k+1}), (1 + 2\sqrt{\mu s})v_{k+1} - v_k \rangle$$

$$- \frac{(1 + \sqrt{\mu s})}{2L} \|\nabla f(x_{k+1}) - \nabla f(x_k)\|^2 - \frac{1}{4} \|v_{k+1} - v_k + \sqrt{s}\nabla f(x_{k+1})\|^2$$

$$\le -\sqrt{\mu s} \left[ \|v_{k+1}\|^2 + \frac{1}{4} (1 + \sqrt{\mu s}) (f(x_{k+1}) - f(x^\star)) + \frac{\mu}{2} \|x_{k+1} - x^\star\|^2 \right]$$

$$- \frac{(1 + \sqrt{\mu s})}{4} \left[ 3\sqrt{\mu s} (f(x_{k+1}) - f(x^\star)) - 2s \|\nabla f(x_{k+1})\|^2 \right]$$

Since $\mu \le L$, then the step size $s \le \mu/(16L^2)$ satisfies it. Hence, the proof is complete with some basic calculations.

# C   Technical Analysis and Proofs for Section 4

## C.1   Technical details for numerical scheme of ODE (4.3)

**Proof of Theorem 4.1 (b)**   The Lyapunov function is constructed as

$$\mathcal{E}(k) = s(k+2)(k+3)(f(x_k) - f(x^\star)) + \frac{1}{2} \left\| 2(x_k - x^\star) + (k+1)\sqrt{s} \left( v_k + \sqrt{s}\nabla f(x_k) \right) \right\|^2.$$

With the basic inequality for $f \in \mathcal{F}_L^1(\mathbb{R}^n)$

$$\begin{cases} f(x_{k+1}) - f(x_k) \le \langle \nabla f(x_{k+1}), x_{k+1} - x_k \rangle \\ f(x_{k+1}) - f(x^\star) \le \langle \nabla f(x_{k+1}), x_{k+1} - x^\star \rangle, \end{cases}$$

we can calculate the iterative difference as

$$\mathcal{E}(k+1) - \mathcal{E}(k)$$
$$= s(k+2)(k+3)(f(x_{k+1}) - f(x_k)) + s(2k+6)(f(x_{k+1}) - f(x^\star))$$
$$+ \frac{1}{2} \langle 2(x_{k+1} - x_k) - \sqrt{s}(k+1)\left(v_k + \sqrt{s}\nabla f(x_k)\right)$$
$$+ \sqrt{s}(k+2)\left(v_{k+1} + \sqrt{s}\nabla f(x_{k+1})\right),$$
$$2(x_{k+1} + x_k - 2x^\star) + \sqrt{s}(k+1)\left(v_k + \sqrt{s}\nabla f(x_k)\right)$$
$$+ \sqrt{s}(k+2)\left(v_{k+1} + \sqrt{s}\nabla f(x_{k+1})\right) \rangle$$
$$= s(k+2)(k+3)(f(x_{k+1}) - f(x_k)) + s(2k+6)(f(x_{k+1}) - f(x^\star))$$
$$- \langle s(k+3)\nabla f(x_{k+1}),$$
$$2(x_{k+1} - x^\star) + \sqrt{s}(k+2)\left(v_{k+1} + \sqrt{s}\nabla f(x_{k+1})\right) \rangle$$
$$- \frac{s^2}{2}(k+3)^2 \|\nabla f(x_{k+1})\|^2$$

$$\leq -\frac{s^2}{2}(k+3)(3k+7)\left\|\nabla f(x_{k+1})\right\|^2.$$

479　Hence, the proof is complete with some basic calculations.

480　**Technical analysis of explicit Euler of ODE** (4.3)　The Lyapunov function is

$$\mathcal{E}(k) = s(k-2)(k+1)(f(x_k)-f(x^\star)) + \frac{1}{2}\left\|2(x_k-x^\star)+(k-1)\sqrt{s}\left(v_k+\sqrt{s}\nabla f(x_k)\right)\right\|^2.$$

481　Then we calculate the iterative difference as

$$
\begin{aligned}
&\mathcal{E}(k+1)-\mathcal{E}(k) \\
&= s(k-1)(k+2)(f(x_{k+1})-f(x_k)) + 2sk\left(f(x_k)-f(x^\star)\right) \\
&\quad +\frac{1}{2}\langle 2(x_{k+1}-x_k)+\sqrt{s}k\left(v_{k+1}+\sqrt{s}\nabla f(x_{k+1})\right) \\
&\qquad\qquad\qquad -\sqrt{s}(k-1)\left(v_k+\sqrt{s}\nabla f(x_k)\right), \\
&\qquad\quad 2(x_{k+1}+x_k-2x^\star)+\sqrt{s}k\left(v_{k+1}+\sqrt{s}\nabla f(x_{k+1})\right) \\
&\qquad\qquad\qquad +\sqrt{s}(k-1)\left(v_k+\sqrt{s}\nabla f(x_k)\right)\rangle \\
&= s(k-1)(k+2)(f(x_{k+1})-f(x_k)) + 2sk\left(f(x_k)-f(x^\star)\right) \\
&\quad -\left\langle s(k+2)\nabla f(x_k), 2(x_k-x^\star)+\sqrt{s}(k-1)\left(v_k+\sqrt{s}\nabla f(x_k)\right)\right\rangle \\
&\quad +\frac{s^2}{2}(k+2)^2\left\|\nabla f(x_k)\right\|^2.
\end{aligned}
$$

482　　　● If we take the following basic inequality for $f \in \mathcal{F}_L^1(\mathbb{R}^n)$

$$
\begin{cases}
f(x_{k+1})-f(x_k) \leq \langle\nabla f(x_k), x_{k+1}-x_k\rangle + \dfrac{L}{2}\|x_{k+1}-x_k\|^2 \\
f(x_k)-f(x^\star) \leq \langle\nabla f(x_k), x_k-x^\star\rangle,
\end{cases}
$$

483　　　we can obtain the following estimate

$$
\begin{aligned}
&\mathcal{E}(k+1)-\mathcal{E}(k) \\
&\leq \frac{Ls^2}{2}(k-1)(k+2)\|v_k\|^2 - 4s\left(f(x_k)-f(x^\star)\right) - \frac{s^2}{2}(k+2)(k-4)\left\|\nabla f(x_k)\right\|^2,
\end{aligned}
$$

484　　　which cannot guarantee the right-hand side of the inequality non-positive.

485　　　● If we take the following basic inequality for $f \in \mathcal{F}_L^1(\mathbb{R}^n)$

$$
\begin{cases}
f(x_{k+1})-f(x_k) \leq \langle\nabla f(x_{k+1}), x_{k+1}-x_k\rangle - \dfrac{1}{2L}\|\nabla f(x_{k+1})-\nabla f(x_k)\|^2 \\
f(x_k)-f(x^\star) \leq \langle\nabla f(x_k), x_k-x^\star\rangle,
\end{cases}
$$

486　　　we can obtain the following estimate

$$
\begin{aligned}
&\mathcal{E}(k+1)-\mathcal{E}(k) \\
&\leq \frac{Ls(k-1)(k+2)}{2}\Bigg(\langle\nabla f(x_{k+1})-\nabla f(x_k), x_{k+1}-x_k\rangle \\
&\qquad\qquad\qquad -\frac{1}{2L}\|\nabla f(x_{k+1})-\nabla f(x_k)\|^2\,t\Bigg) \\
&\quad -4s\left(f(x_k)-f(x^\star)\right) - \frac{s^2}{2}(k+2)(k-4)\left\|\nabla f(x_k)\right\|^2,
\end{aligned}
$$

487　　　which cannot guarantee the right-hand side of the inequality non-positive.

 ## C.2 Technical details for standard numerical schemes

489 Standard Euler discretization of ODE (4.1), with initial $x_0$ and $v_0 = -\sqrt{s}\nabla f(x_0)$, are shown as
490 below. **Euler scheme of** (4.1)**: (S), (E) and (I) respectively**

**(S)**
$$\begin{cases} x_{k+1} - x_k = \sqrt{s}v_k \\ v_{k+1} - v_k = -\dfrac{3v_{k+1}}{k+1} - \sqrt{s}\left(\nabla f(x_{k+1}) - \nabla f(x_k)\right) - \sqrt{s}\left(\dfrac{2k+5}{2k+2}\right)\nabla f(x_{k+1}). \end{cases}$$

**(E)**
$$\begin{cases} x_{k+1} - x_k = \sqrt{s}v_k \\ v_{k+1} - v_k = -\dfrac{3v_k}{k} - \sqrt{s}\left(\nabla f(x_{k+1}) - \nabla f(x_k)\right) - \sqrt{s}\left(\dfrac{2k+3}{2k}\right)\nabla f(x_k). \end{cases}$$

**(I)**
$$\begin{cases} x_{k+1} - x_k = \sqrt{s}v_{k+1} \\ v_{k+1} - v_k = -\dfrac{3v_{k+1}}{k+1} - \sqrt{s}\left(\nabla f(x_{k+1}) - \nabla f(x_k)\right) - \sqrt{s}\left(\dfrac{2k+5}{2k+2}\right)\nabla f(x_{k+1}). \end{cases}$$

491 **Technical analysis of symplectic scheme of ODE** (4.1)   The Lyapunov function is

$$\mathcal{E}(k) = s(k+1)\left(k+\frac{3}{2}\right)(f(x_k) - f(x^\star)) + \frac{1}{2}\left\|2(x_{k+1} - x^\star) + (k+1)\sqrt{s}\left(v_k + \sqrt{s}\nabla f(x_k)\right)\right\|^2.$$

492 Then we calculate the iterative difference as

$$\mathcal{E}(k+1) - \mathcal{E}(k)$$
$$= s(k+1)\left(k+\frac{3}{2}\right)(f(x_{k+1}) - f(x_k)) + s\left(2k+\frac{7}{2}\right)(f(x_{k+1}) - f(x^\star))$$
$$+ \frac{1}{2}\langle 2(x_{k+2} - x_{k+1}) - \sqrt{s}(k+1)\left(v_k + \sqrt{s}\nabla f(x_k)\right)$$
$$+ \sqrt{s}(k+2)\left(v_{k+1} + \sqrt{s}\nabla f(x_{k+1})\right)$$
$$2(x_{k+2} + x_{k+1} - 2x^\star) + \sqrt{s}(k+1)\left(v_k + \sqrt{s}\nabla f(x_k)\right)$$
$$+ \sqrt{s}(k+2)\left(v_{k+1} + \sqrt{s}\nabla f(x_{k+1})\right)\rangle$$
$$= s(k+1)\left(k+\frac{3}{2}\right)(f(x_{k+1}) - f(x_k)) + s\left(2k+\frac{7}{2}\right)(f(x_{k+1}) - f(x^\star))$$
$$- \left\langle s\left(k+\frac{3}{2}\right)\nabla f(x_{k+1}), 2(x_{k+2} - x^\star) + \sqrt{s}(k+2)\left(v_{k+1} + \sqrt{s}\nabla f(x_{k+1})\right)\right\rangle$$
$$- \frac{s^2}{2}\left(k+\frac{3}{2}\right)^2\|\nabla f(x_{k+1})\|^2.$$

493 Now we hope to utilize the basic inequality for $f \in \mathcal{F}_L^1(\mathbb{R}^n)$ to make the right side of equality no
494 more than zero. Taking the following inequalities

$$\begin{cases} f(x_{k+1}) - f(x_k) \leq \langle \nabla f(x_{k+1}), x_{k+1} - x_k\rangle - \dfrac{1}{2L}\|\nabla f(x_{k+1}) - \nabla f(x_k)\|^2 \\ f(x_{k+1}) - f(x^\star) \leq \langle \nabla f(x_{k+1}), x_{k+1} - x^\star\rangle, \end{cases}$$

495 we can obtain the iterative difference is

$$\mathcal{E}(k+1) - \mathcal{E}(k)$$
$$\leq \frac{s}{2}(f(x_{k+1}) - f(x^\star)) - \frac{s^2}{2}\left(k+\frac{3}{2}\right)\left(3k+\frac{7}{2}\right)\|\nabla f(x_{k+1})\|^2$$
$$- \frac{s}{2L}(k+1)\left(k+\frac{3}{2}\right)\|\nabla f(x_{k+1}) - \nabla f(x_k)\|^2$$
$$+ s^2(k+1)\left(k+\frac{3}{2}\right)\langle \nabla f(x_{k+1}), \nabla f(x_{k+1}) - \nabla f(x_k)\rangle$$

$$+s^2 \left(k + \frac{3}{2}\right)\left(k + \frac{5}{2}\right) \|\nabla f(x_{k+1})\|^2$$

$$\leq \frac{s}{2}\left(f(x_{k+1}) - f(x^\star)\right) - \frac{s^2}{2}\left(k + \frac{3}{2}\right)\left(k - \frac{3}{2} - Ls(k+1)\right)\|\nabla f(x_{k+1})\|^2.$$

Since there exists a non-negative term, $\frac{s}{2}\left(f(x_{k+1}) - f(x^\star)\right)$, we cannot guarantee the right-hand side of inequality is non-positive. Hence, the convergence cannot be proved by the above description.

**Technical analysis of explicit scheme of ODE** (4.1)    The Lyapunov function is

$$\mathcal{E}(k) = s(k-2)\left(k - \frac{1}{2}\right)\left(f(x_k) - f(x^\star)\right) + \frac{1}{2}\left\|2(x_k - x^\star) + (k-1)\sqrt{s}\left(v_k + \sqrt{s}\nabla f(x_k)\right)\right\|^2.$$

Then we calculate the iterative difference as

$$\mathcal{E}(k+1) - \mathcal{E}(k)$$

$$= s(k-1)\left(k + \frac{1}{2}\right)\left(f(x_{k+1}) - f(x_k)\right) + s\left(2k - \frac{3}{2}\right)\left(f(x_k) - f(x^\star)\right)$$

$$+ \frac{1}{2}\langle 2(x_{k+1} - x_k) + \sqrt{s}k\left(v_{k+1} + \sqrt{s}\nabla f(x_{k+1})\right)$$

$$- \sqrt{s}(k-1)\left(v_k + \sqrt{s}\nabla f(x_k)\right),$$

$$2(x_{k+1} + x_k - 2x^\star) + \sqrt{s}k\left(v_{k+1} + \sqrt{s}\nabla f(x_{k+1})\right)$$

$$+ \sqrt{s}(k-1)\left(v_k + \sqrt{s}\nabla f(x_k)\right)\rangle$$

$$= s(k-1)\left(k + \frac{1}{2}\right)\left(f(x_{k+1}) - f(x_k)\right) + s\left(2k - \frac{3}{2}\right)\left(f(x_k) - f(x^\star)\right)$$

$$- \left\langle s\left(k + \frac{1}{2}\right)\nabla f(x_k), 2(x_k - x^\star) + \sqrt{s}(k-1)\left(v_k + \sqrt{s}\nabla f(x_k)\right)\right\rangle$$

$$+ \frac{s^2}{2}\left(k + \frac{1}{2}\right)^2 \|\nabla f(x_k)\|^2.$$

- If we take the following basic inequality for $f \in \mathcal{F}_L^1(\mathbb{R}^n)$

$$\begin{cases} f(x_{k+1}) - f(x_k) \leq \langle \nabla f(x_k), x_{k+1} - x_k\rangle + \frac{L}{2}\|x_{k+1} - x_k\|^2 \\ f(x_k) - f(x^\star) \leq \langle \nabla f(x_k), x_k - x^\star\rangle, \end{cases}$$

we can obtain the following estimate

$$\mathcal{E}(k+1) - \mathcal{E}(k) \leq \frac{Ls^2}{2}(k-1)\left(k + \frac{1}{2}\right)\|v_k\|^2$$

$$- \frac{5s}{2}\left(f(x_k) - f(x^\star)\right) - \frac{s^2}{2}\left(k + \frac{1}{2}\right)\left(k - \frac{5}{2}\right)\|\nabla f(x_k)\|^2,$$

which cannot guarantee the right-hand side of the inequality non-positive.

- If we take the following basic inequality for $f \in \mathcal{F}_L^1(\mathbb{R}^n)$

$$\begin{cases} f(x_{k+1}) - f(x_k) \leq \langle \nabla f(x_{k+1}), x_{k+1} - x_k\rangle - \frac{1}{2L}\|\nabla f(x_{k+1}) - \nabla f(x_k)\|^2 \\ f(x_k) - f(x^\star) \leq \langle \nabla f(x_k), x_k - x^\star\rangle, \end{cases}$$

we can obtain the following estimate

$$\mathcal{E}(k+1) - \mathcal{E}(k)$$

$$\leq \frac{Ls(k-1)(2k+1)}{4}\Bigg(\langle \nabla f(x_{k+1}) - \nabla f(x_k), x_{k+1} - x_k\rangle$$

$$- \frac{1}{2L}\|\nabla f(x_{k+1}) - \nabla f(x_k)\|^2\Bigg)$$

$$-\frac{5s}{2}\left(f(x_k)-f(x^\star)\right)-\frac{s^2}{2}\left(k+\frac{1}{2}\right)\left(k-\frac{5}{2}\right)\|\nabla f(x_k)\|^2,$$

which cannot guarantee the right-hand side of the inequality non-positive.

**Technical analysis of implicit scheme of ODE** (4.1)  The Lyapunov function is

$$\mathcal{E}(k)=s\,(k+2)\left(k+\frac{3}{2}\right)(f(x_k)-f(x^\star))$$

$$+\frac{1}{2}\left\|2(x_k-x^\star)+(k+1)\sqrt{s}\left(v_k+\sqrt{s}\nabla f(x_k)\right)\right\|^2.$$

Then we can calculate the iterative difference as

$$\mathcal{E}(k+1)-\mathcal{E}(k)$$

$$=s\,(k+2)\left(k+\frac{3}{2}\right)(f(x_{k+1})-f(x_k))+s\left(2k+\frac{9}{2}\right)(f(x_{k+1})-f(x^\star))$$

$$+\frac{1}{2}\big\langle 2(x_{k+1}-x_k)-\sqrt{s}(k+1)\left(v_k+\sqrt{s}\nabla f(x_k)\right)$$

$$+\sqrt{s}(k+2)\left(v_{k+1}+\sqrt{s}\nabla f(x_{k+1})\right),$$

$$2(x_{k+1}+x_k-2x^\star)+\sqrt{s}(k+1)\left(v_k+\sqrt{s}\nabla f(x_k)\right)$$

$$+\sqrt{s}(k+2)\left(v_{k+1}+\sqrt{s}\nabla f(x_{k+1})\right)\big\rangle$$

$$=s\,(k+2)\left(k+\frac{3}{2}\right)(f(x_{k+1})-f(x_k))+s\left(2k+\frac{9}{2}\right)(f(x_{k+1})-f(x^\star))$$

$$-\left\langle s\left(k+\frac{3}{2}\right)\nabla f(x_{k+1}),2(x_{k+1}-x^\star)+\sqrt{s}(k+2)\left(v_{k+1}+\sqrt{s}\nabla f(x_{k+1})\right)\right\rangle$$

$$-\frac{s^2}{2}\left(k+\frac{3}{2}\right)^2\|\nabla f(x_{k+1})\|^2.$$

Now we hope to utlize the basic inequality for $f\in\mathcal{F}_L^1(\mathbb{R}^n)$ to make the right side of equality no more than zero. Taking the following inequalities

$$\begin{cases}f(x_{k+1})-f(x_k)\le\langle\nabla f(x_{k+1}),x_{k+1}-x_k\rangle\\ f(x_{k+1})-f(x^\star)\le\langle\nabla f(x_{k+1}),x_{k+1}-x^\star\rangle,\end{cases}$$

we can obtain

$$\mathcal{E}(k+1)-\mathcal{E}(k)\le\frac{3s}{2}\left(f(x_{k+1})-f(x^\star)\right)-\frac{s^2}{2}\left(k+\frac{3}{2}\right)\left(3k+\frac{7}{2}\right)\|\nabla f(x_{k+1})\|^2.$$

Although the negative term concludes the multiplier $k^2$, we cannot guarantee the right-hand side non-positive

# D  Low-Resolution ODEs

## D.1  Low-resolution ODE for strongly convex functions

In this subsection, we discuss the numerical discretization of (2.3). We rewrite this ODE in a phase-space representation

$$\begin{cases}\dot{X}=V\\ \dot{V}=-2\sqrt{\mu}V-\nabla f(X)\end{cases},\tag{D.1}$$

with $X(0)=x_0$ and $V(0)=0$. We have the following theorem:

**Theorem D.1.** Let $f\in\mathcal{S}_{\mu,L}^1(\mathbb{R}^n)$. The solution $X=X(t)$ to low-resolution ODE (2.3) satisfies

$$f(X)-f(x^\star)\le\frac{3L\,\|x_0-x^\star\|^2}{2}\mathrm{e}^{-\frac{\sqrt{\mu}t}{4}}.\tag{D.2}$$

519   *Proof.* The Lyapunov function is

$$\mathcal{E} = \frac{1}{4}\|\dot{X}\|^2 + \frac{1}{4}\|2\sqrt{\mu}(X - x^\star) + \dot{X}\|^2 + f(X) - f(x^\star).$$

520   Using the Cauchy-Schwartz inequality

$$\|2\sqrt{\mu}(X - x^\star) + \dot{X}\|^2 \le 2\left(4\mu\|X - x^\star\|^2 + \|\dot{X}\|^2\right),$$

521   and the basic inequality for $f \in \mathcal{S}^1_{\mu,L}(\mathbb{R}^n)$

$$f(x^\star) \ge f(X) + \langle \nabla f(X), x^\star - x \rangle + \frac{\mu}{2}\|X - x^\star\|^2,$$

522   we calculate the time derivative

$$\begin{aligned}
\frac{d\mathcal{E}}{dt} &= \frac{1}{2}\left\langle \dot{X}, -2\sqrt{\mu}\dot{X} - \nabla f(X) \right\rangle + \frac{1}{2}\left\langle 2\sqrt{\mu}(X - x^\star) + \dot{X}, -\nabla f(X) \right\rangle + \left\langle \nabla f(X), \dot{X} \right\rangle \\
&= -\sqrt{\mu}\left(\|\dot{X}\|^2 + \langle \nabla f(X), X - x^\star \rangle\right) \\
&\le -\sqrt{\mu}\left(\|\dot{X}\|^2 + f(X) - f(x^\star) + \frac{\mu}{2}\|X - x^\star\|^2\right) \\
&\le -\frac{\sqrt{\mu}}{4}\mathcal{E}.
\end{aligned}$$

523   Hence, the proof is complete.     $\square$

524   We now analyze the standard Euler discretization of the low-resolution ODE (2.3). All of the
525   following three Euler schemes take the same initial $x_0$ and $v_0 = 0$.

526   **Euler Scheme of ODE** (2.3)**: (S), (E) and (I) respectively**

$$\textbf{(S)} \qquad \begin{cases} x_{k+1} - x_k = \sqrt{s}v_k \\ v_{k+1} - v_k = -2\sqrt{\mu s}v_{k+1} - \sqrt{s}\nabla f(x_{k+1}). \end{cases}$$

$$\textbf{(E)} \qquad \begin{cases} x_{k+1} - x_k = \sqrt{s}v_k \\ v_{k+1} - v_k = -2\sqrt{\mu s}v_k - \sqrt{s}\nabla f(x_k). \end{cases}$$

$$\textbf{(I)} \qquad \begin{cases} x_{k+1} - x_k = \sqrt{s}v_{k+1} \\ v_{k+1} - v_k = -2\sqrt{\mu s}v_{k+1} - \sqrt{s}\nabla f(x_{k+1}). \end{cases}$$

527   **Theorem D.2** (Discretization of Low-Resolution ODE — General)**.** For any $f \in \mathcal{S}^1_{\mu,L}(\mathbb{R}^n)$, the
528   following conclusions hold:

529     (a) Taking $0 < s \le \mu/(16L^2)$, the symplectic Euler scheme satisfies

$$f(x_k) - f(x^\star) \le \frac{3L\|x_0 - x^\star\|^2}{2\left(1 + \frac{\sqrt{\mu s}}{4}\right)^k}. \tag{D.3}$$

530     (b) Taking $0 < s \le \mu/(25L^2)$, the explicit Euler scheme satisfies

$$f(x_k) - f(x^\star) \le \frac{3L\|x_0 - x^\star\|^2}{2}\left(1 - \frac{\sqrt{\mu s}}{8}\right)^k. \tag{D.4}$$

531     (c) Taking $0 < s \le 1/L$, the implicit Euler scheme satisfies

$$f(x_k) - f(x^\star) \le \frac{3L\|x_0 - x^\star\|^2}{2\left(1 + \frac{\sqrt{\mu s}}{4}\right)^k}. \tag{D.5}$$

532 *Proof.*  (a) The Lyapunov function is

$$\mathcal{E}(k) = \frac{1}{4}\left\|v_k\right\|^2 + \frac{1}{4}\left\|2\sqrt{\mu}(x_k - x^\star) + v_k\right\|^2 + f(x_k) - f(x^\star).$$

533 With the Cauchy-Schwartz inequality

$$\left\|2\sqrt{\mu}(x_k - x^\star) + v_k\right\|^2 \le 2\left(4\mu\left\|x_k - x^\star\right\|^2 + \left\|v_k\right\|^2\right),$$

534 and the basic inequality for $f \in \mathcal{S}_{\mu,L}^1(\mathbb{R}^n)$

$$\begin{cases} f(x_{k+1}) - f(x_k) \le \langle \nabla f(x_{k+1}), x_{k+1} - x_k \rangle \\ f(x^\star) \ge f(x_{k+1}) + \langle \nabla f(x_{k+1}), x^\star - x_{k+1} \rangle + \frac{\mu}{2}\left\|x_{k+1} - x^\star\right\|^2, \end{cases}$$

535 we calculate the iterave difference

$$\mathcal{E}(k+1) - \mathcal{E}(k)$$
$$= \frac{1}{4}\left\langle v_{k+1} - v_k, v_{k+1} + v_k \right\rangle + f(x_{k+1}) - f(x_k)$$
$$\quad + \frac{1}{4}\left\langle 2\sqrt{\mu}(x_{k+1} - x_k) + v_{k+1} - v_k, 2\sqrt{\mu}(x_{k+1} + x_k - 2x^\star) + v_{k+1} + v_k \right\rangle$$
$$\le \frac{1}{2}\left\langle v_{k+1} - v_k, v_{k+1} \right\rangle + \langle \nabla f(x_{k+1}), x_{k+1} - x_k \rangle$$
$$\quad + \frac{1}{2}\left\langle 2\sqrt{\mu}(x_{k+1} - x_k) + v_{k+1} - v_k, 2\sqrt{\mu}(x_{k+1} - x^\star) + v_{k+1} \right\rangle$$
$$\quad - \frac{1}{4}\left\|v_{k+1} - v_k\right\|^2 - \frac{1}{4}\left\|2\sqrt{\mu}(x_{k+1} - x_k) + v_{k+1} - v_k\right\|^2$$
$$\le -\sqrt{\mu s}\left(\left\|v_{k+1}\right\|^2 + \langle \nabla f(x_{k+1}), x_{k+1} - x^\star \rangle\right)$$
$$\le -\sqrt{\mu s}\left(\left\|v_{k+1}\right\|^2 + f(x_{k+1}) - f(x^\star) + \frac{\mu}{2}\left\|x_{k+1} - x^\star\right\|^2\right)$$
$$\le -\frac{\sqrt{\mu s}}{4}\mathcal{E}(k+1).$$

536 Hence, the proof is complete.

537 (b) The Lyapunov function is

$$\mathcal{E}(k) = \frac{1}{4}\left\|v_k\right\|^2 + \frac{1}{4}\left\|2\sqrt{\mu}(x_k - x^\star) + v_k\right\|^2 + f(x_k) - f(x^\star).$$

538 With the Cauchy-Schwartz inequality

$$\left\|2\sqrt{\mu}(x_k - x^\star) + v_k\right\|^2 \le 2\left(4\mu\left\|x_k - x^\star\right\|^2 + \left\|v_k\right\|^2\right),$$

539 and the basic inequality for $f \in \mathcal{S}_{\mu,L}^1(\mathbb{R}^n)$

$$\begin{cases} f(x_{k+1}) - f(x_k) \le \langle \nabla f(x_k), x_{k+1} - x_k \rangle + \frac{L}{2}\left\|x_{k+1} - x_k\right\|^2 \\ f(x^\star) \ge f(x_{k+1}) + \langle \nabla f(x_{k+1}), x^\star - x_{k+1} \rangle + \frac{\mu}{2}\left\|x^\star - x_{k+1}\right\|^2, \end{cases}$$

540 we calculate the iterave difference

$$\mathcal{E}(k+1) - \mathcal{E}(k)$$
$$= \frac{1}{4}\left\langle v_{k+1} - v_k, v_{k+1} + v_k \right\rangle + f(x_{k+1}) - f(x_k)$$
$$\quad + \frac{1}{4}\left\langle 2\sqrt{\mu}(x_{k+1} - x_k) + v_{k+1} - v_k, 2\sqrt{\mu}(x_{k+1} + x_k - 2x^\star) + v_{k+1} + v_k \right\rangle$$
$$\le \frac{1}{2}\left\langle v_{k+1} - v_k, v_k \right\rangle + \langle \nabla f(x_k), x_{k+1} - x_k \rangle + \frac{L}{2}\left\|x_{k+1} - x_k\right\|^2$$

$$+\frac{1}{2}\left\langle 2\sqrt{\mu}(x_{k+1}-x_k)+v_{k+1}-v_k, 2\sqrt{\mu}(x_k-x^\star)+v_k\right\rangle$$

$$+\frac{1}{4}\left\|v_{k+1}-v_k\right\|^2+\frac{1}{4}\left\|2\sqrt{\mu}(x_{k+1}-x_k)+v_{k+1}-v_k\right\|^2$$

$$\leq -\sqrt{\mu s}\left(\left\|v_k\right\|^2+\left\langle\nabla f(x_k), x_k-x^\star\right\rangle\right)$$

$$+\frac{Ls}{2}\left\|v_k\right\|^2+\frac{s}{4}\left\|2\sqrt{\mu}v_k+\nabla f(x_k)\right\|^2+\frac{s}{4}\left\|\nabla f(x_k)\right\|^2$$

$$\leq -\frac{\sqrt{\mu s}}{2}\left(\left\|v_k\right\|^2+f(x_k)-f(x^\star)+\frac{\mu}{2}\left\|x_k-x^\star\right\|^2\right)$$

$$-\frac{\sqrt{\mu s}}{2}\left(\left\|v_k\right\|^2+\frac{1}{L}\left\|\nabla f(x_k)\right\|^2\right)+s\left(2\mu+\frac{L}{2}\right)\left\|v_k\right\|^2+\frac{3s}{4}\left\|\nabla f(x_k)\right\|^2.$$

Since $\mu\leq L$, the step size $s\leq\mu/(25L^2)$ satisfies it. Hence, the proof is complete after some basic calculations.

(c) The Lyapunov function is

$$\mathcal{E}(k)=\frac{1}{4}\left\|v_k\right\|^2+\frac{1}{4}\left\|2\sqrt{\mu}(x_{k+1}-x^\star)+v_k\right\|^2+f(x_k)-f(x^\star).$$

With the Cauchy-Schwartz inequality

$$\left\|2\sqrt{\mu}(x_{k+1}-x^\star)+v_k\right\|^2=\left\|2\sqrt{\mu}(x_k-x^\star)+(1+2\sqrt{\mu s})v_k\right\|^2$$

$$\leq 2\left(4\mu\left\|x_k-x^\star\right\|^2+(1+2\sqrt{\mu s})^2\left\|v_k\right\|^2\right),$$

and the basic inequality for $f\in\mathcal{S}_{\mu,L}^1(\mathbb{R}^n)$

$$\begin{cases} f(x_{k+1})-f(x_k)\leq\left\langle\nabla f(x_{k+1}), x_{k+1}-x_k\right\rangle-\frac{1}{2L}\left\|\nabla f(x_{k+1})-\nabla f(x_k)\right\|^2 \\ f(x^\star)\geq f(x_{k+1})+\left\langle\nabla f(x_{k+1}), x^\star-x_{k+1}\right\rangle+\frac{\mu}{2}\left\|x^\star-x_{k+1}\right\|^2, \end{cases}$$

we calculate the iterave difference

$$\mathcal{E}(k+1)-\mathcal{E}(k)$$

$$=\frac{1}{4}\left\langle v_{k+1}-v_k, v_{k+1}+v_k\right\rangle+f(x_{k+1})-f(x_k)$$

$$+\frac{1}{4}\left\langle 2\sqrt{\mu}(x_{k+2}-x_{k+1})+v_{k+1}-v_k, 2\sqrt{\mu}(x_{k+2}+x_{k+1}-2x^\star)+v_{k+1}+v_k\right\rangle$$

$$\leq\frac{1}{2}\left\langle v_{k+1}-v_k, v_{k+1}\right\rangle-\frac{1}{4}\left\|v_{k+1}-v_k\right\|^2$$

$$+\left\langle\nabla f(x_{k+1}), x_{k+1}-x_k\right\rangle-\frac{1}{2L}\left\|\nabla f(x_{k+1})-\nabla f(x_k)\right\|^2$$

$$+\frac{1}{2}\left\langle-\sqrt{s}\nabla f(x_{k+1}), 2\sqrt{\mu}(x_{k+2}-x^\star)+v_{k+1}\right\rangle-\frac{1}{4}\left\|\sqrt{s}\nabla f(x_{k+1})\right\|^2$$

$$\leq-\sqrt{\mu s}\left(\left\|v_{k+1}\right\|^2+\left\langle\nabla f(x_{k+1}), x_{k+1}-x^\star\right\rangle\right)$$

$$-\frac{\sqrt{s}}{2}\left\langle\nabla f(x_{k+1}), (1+2\sqrt{\mu s})v_{k+1}-v_k\right\rangle$$

$$-\frac{1}{2L}\left\|\nabla f(x_{k+1})-\nabla f(x_k)\right\|^2-\frac{1}{4}\left\|v_{k+1}-v_k+\sqrt{s}\nabla f(x_{k+1})\right\|^2$$

$$\leq-\sqrt{\mu s}\left[\left\|v_{k+1}\right\|^2+\frac{1}{4}\left(f(x_{k+1})-f(x^\star)\right)+\frac{\mu}{2}\left\|x_{k+1}-x^\star\right\|^2\right]$$

$$-\frac{1}{4}\left[3\sqrt{\mu s}\left(f(x_{k+1})-f(x^\star)\right)-2s\left\|\nabla f(x_{k+1})\right\|^2\right].$$

Since $\mu\leq L$, the step size $s\leq\mu/(16L^2)$ satisfies it. Hence, the proof is complete after some basic calculations.

$$\square$$

**Corollary D.3** (Discretization of NAG-SC low-resolution ODE). For any $f \in \mathcal{S}_{\mu,L}^1(\mathbb{R}^n)$, the following conclusions hold:

(a) Taking step size $0s = \mu/(16L^2)$ , the symplectic Euler scheme satisfies

$$f(x_k) - f(x^\star) \leq \frac{3L \|x_0 - x^\star\|^2}{2 \left(1 + \frac{\mu}{16L}\right)^k}. \tag{D.6}$$

(b) Taking step size $s = \mu/(16L^2)$, the explicit Euler scheme satisfies

$$f(x_k) - f(x^\star) \leq \frac{3L \|x_0 - x^\star\|^2}{2} \left(1 - \frac{\mu}{40L}\right)^k. \tag{D.7}$$

(c) Taking step size $s = 1/L$, the implicit Euler scheme satisfies

$$f(x_k) - f(x^\star) \leq \frac{3L \|x_0 - x^\star\|^2}{2 \left(1 + \frac{1}{4}\sqrt{\frac{\mu}{L}}\right)^k}. \tag{D.8}$$

**Remark D.1.** Compared with Theorem D.2 (a) – (c), just the Euler scheme of the low-resolution ODE (2.3), both the explicit scheme and the symplectic scheme can retain the convergence rate from the continuous version of Theorem D.1, when the step size $s$ is of the order $O(\mu/L^2)$. Although the explicit scheme is weaker than the symplectic scheme, it can preserve the rate to the same order as the symplectic scheme. However, if the step size satisfies $s = O(\mu/L^2)$, the algorithm cannot provide acceleration. There is no limitation on the step size $s$ for the implicit Euler scheme, but in general it is not practical for non-quadratic objective functions.

## D.2 Low-resolution ODE for convex functions

In this subsection, we discuss the numerical discretization of (2.2). We rewrite it in a phase-space representation:

$$\begin{cases} \dot{X} = V \\ \dot{V} = -\dfrac{3}{t}V - \nabla f(X), \end{cases} \tag{D.9}$$

with $X(0) = x_0$ and $V(0) = 0$.

**Theorem D.4.** Let $f \in \mathcal{F}_L^1(\mathbb{R}^n)$. The solution $X = X(t)$ to the low-resolution ODE (2.2) satisfies

$$\begin{cases} f(X) - f(x^\star) \leq \dfrac{2 \|x_0 - x^\star\|^2}{t^2} \\ \min\limits_{0 \leq u \leq t} \|\nabla f(X(u))\|^2 \leq \dfrac{4L \|x_0 - x^\star\|^2}{t^2}. \end{cases} \tag{D.10}$$

Theorem D.4 is combined with Theorem 3 Su et al. [2016] and a further analysis about gradient norm minimization in Shi et al. [2018]. The Lyapunov function is constructed in Su et al. [2016] as

$$\mathcal{E} = t^2 \left(f(X) - f(x^\star)\right) + \frac{1}{2}\|2(X - x^\star) + t\dot{X}\|^2. \tag{D.11}$$

### D.2.1 Symplectic Euler scheme

First, we utilize the symplectic Euler scheme with the initial $x_0$ and $v_0 = 0$, as shown as following:

$$\begin{cases} x_{k+1} - x_k = \sqrt{s}v_k \\ v_{k+1} - v_k = -\dfrac{3}{k+1}v_{k+1} - \sqrt{s}\nabla f(x_{k+1}). \end{cases} \tag{D.12}$$

571 **Technical analysis of symplectic scheme** (D.12)    The Lyapunov function is

$$\mathcal{E}(k) = (k+1)^2 s \left( f(x_k) - f(x^\star) \right) + \frac{1}{2} \left\| 2(x_{k+1} - x^\star) + (k+1)\sqrt{s}v_k \right\|^2.$$

572 Then we can calculate the iterate difference as

$$
\begin{aligned}
\mathcal{E}(k+1) &- \mathcal{E}(k) \\
&= (k+1)^2 s \left( f(x_{k+1}) - f(x_k) \right) + (2k+3)s \left( f(x_{k+1}) - f(x^\star) \right) \\
&\quad + \frac{1}{2} \big\langle 2(x_{k+1} - x_k) + (k+2)\sqrt{s}v_{k+1} - (k+1)\sqrt{s}v_k, \\
&\qquad\qquad 2(x_{k+1} + x_k - 2x^\star) + (k+2)\sqrt{s}v_{k+1} + (k+1)\sqrt{s}v_k \big\rangle \\
&= (k+1)^2 s \left( f(x_{k+1}) - f(x_k) \right) + (2k+3)s \left( f(x_{k+1}) - f(x^\star) \right) \\
&\quad - \big\langle (k+1)s\nabla f(x_{k+1}), 2(x_{k+2} - x^\star) + (k+2)\sqrt{s}v_{k+1} \big\rangle \\
&\quad - \frac{1}{2}(k+1)^2 s^2 \left\| \nabla f(x_{k+1}) \right\|^2.
\end{aligned}
$$

573 We hope to utilize the basic inequality for $f \in \mathcal{F}_L^1(\mathbb{R}^n)$ to make the right-hand-side of the equality
574 no more than zero. Based on the following inequalities:

$$
\begin{cases}
f(x_{k+1}) - f(x_k) \leq \langle \nabla f(x_{k+1}), x_{k+1} - x_k \rangle - \dfrac{1}{2L} \left\| \nabla f(x_{k+1}) - \nabla f(x_k) \right\|^2 \\[2mm]
f(x_{k+1}) - f(x^\star) \leq \langle \nabla f(x_{k+1}), x_{k+1} - x^\star \rangle - \dfrac{1}{2L} \left\| \nabla f(x_{k+1}) \right\|^2,
\end{cases}
$$

575 we obtain the following estimate:

$$\mathcal{E}(k+1) - \mathcal{E}(k) \leq \frac{1}{2}(k+1)^2 s^2 \left\| \nabla f(x_{k+1}) \right\|^2 + s \left( f(x_{k+1}) - f(x^\star) \right) - \frac{(k+1)s}{L} \left\| \nabla f(x_{k+1}) \right\|^2,$$

576 from which we cannot guarantee that the right-hand-side of the inequality is nonpositive.

### D.2.2    Explicit Euler scheme

578 Now, we turn to the explicit Euler scheme with the initial $x_0$ and $v_0 = 0$, as

$$
\begin{cases}
x_{k+1} - x_k = \sqrt{s}v_k \\[2mm]
v_{k+1} - v_k = -\dfrac{3}{k}v_k - \sqrt{s}\nabla f(x_k).
\end{cases}
\tag{D.13}
$$

579 **Technical analysis of explicit scheme** (D.13)    Now, the Lyapunov function is

$$\mathcal{E}(k) = (k-2)(k-1)s \left( f(x_k) - f(x^\star) \right) + \frac{1}{2} \left\| 2(x_k - x^\star) + (k-1)\sqrt{s}v_k \right\|^2.$$

580 Then we can calculate the iterate difference as

$$
\begin{aligned}
\mathcal{E}(k+1) &- \mathcal{E}(k) \\
&= (k-1)ks \left( f(x_{k+1}) - f(x_k) \right) + 2(k-1)s \left( f(x_k) - f(x^\star) \right) \\
&\quad + \frac{1}{2} \big\langle 2(x_{k+1} - x_k) + k\sqrt{s}v_{k+1} - (k-1)\sqrt{s}v_k, \\
&\qquad\qquad 2(x_{k+1} + x_k - 2x^\star) + k\sqrt{s}v_{k+1} + (k-1)\sqrt{s}v_k \big\rangle \\
&\quad (k-1)ks \left( f(x_{k+1}) - f(x_k) \right) + 2(k-1)s \left( f(x_k) - f(x^\star) \right) \\
&\quad + \big\langle -ks\nabla f(x_k), 2(x_k - x^\star) + (k-1)\sqrt{s}v_k \big\rangle + \frac{1}{2}k^2 s^2 \left\| \nabla f(x_k) \right\|^2.
\end{aligned}
$$

581 - If we take the following basic inequality for $f \in \mathcal{F}_L^1(\mathbb{R}^n)$

$$
\begin{cases}
f(x_{k+1}) - f(x_k) \leq \langle \nabla f(x_k), x_{k+1} - x_k \rangle + \dfrac{L}{2} \left\| x_{k+1} - x_k \right\|^2 \\[2mm]
f(x_k) - f(x^\star) \leq \langle \nabla f(x_k), x_k - x^\star \rangle - \dfrac{1}{2L} \left\| \nabla f(x_k) \right\|^2,
\end{cases}
$$

we obtain the following estimate:

$$\mathcal{E}(k+1) - \mathcal{E}(k) \le \frac{k(k-1)Ls}{2} \left\| x_{k+1} - x_k \right\|^2$$

$$- 2s \left( f(x_k) - f(x^\star) \right) - \frac{ks}{L} \left\| \nabla f(x_k) \right\|^2 + \frac{k^2 s^2}{2} \left\| \nabla f(x_k) \right\|^2,$$

from which we cannot guarantee that the right-hand-side of the inequality is nonpositive.

- If we take the following basic inequality for $f \in \mathcal{F}_L^1(\mathbb{R}^n)$

$$\begin{cases} f(x_{k+1}) - f(x_k) \le \langle \nabla f(x_{k+1}), x_{k+1} - x_k \rangle - \dfrac{1}{2L} \left\| \nabla f(x_{k+1}) - \nabla f(x_k) \right\|^2 \\ f(x_k) - f(x^\star) \le \langle \nabla f(x_k), x_k - x^\star \rangle - \dfrac{1}{2L} \left\| \nabla f(x_k) \right\|^2, \end{cases}$$

we obtain the following estimate:

$$\mathcal{E}(k+1) - \mathcal{E}(k)$$

$$\le (k-1)ks \left( \langle \nabla f(x_{k+1}) - \nabla f(x_k), x_{k+1} - x_k \rangle - \frac{1}{2L} \left\| \nabla f(x_{k+1}) - \nabla f(x_k) \right\|^2 \right)$$

$$- 2s \left( f(x_k) - f(x^\star) \right) - \frac{ks}{L} \left\| \nabla f(x_k) \right\|^2 + \frac{k^2 s^2}{2} \left\| \nabla f(x_k) \right\|^2,$$

from which we still cannot guarantee that the right-hand-side of the inequality is nonpositive.

### D.2.3 Implicit scheme

Finally, we analyze the implicit Euler scheme with the initial $x_0$ and $v_0 = 0$:

$$\begin{cases} x_{k+1} - x_k = \sqrt{s} v_{k+1} \\ v_{k+1} - v_k = -\dfrac{3}{k+1} v_{k+1} - \sqrt{s} \nabla f(x_{k+1}) \end{cases} \tag{D.14}$$

**Technical analysis of implicit scheme** (D.14)   We construct the Lyapunov function as

$$\mathcal{E}(k) = (k+1)(k+2)s \left( f(x_k) - f(x^\star) \right) + \frac{1}{2} \left\| 2(x_k - x^\star) + (k+1)\sqrt{s} v_k \right\|^2.$$

Then we can calculate the iterate difference as

$$\mathcal{E}(k+1) - \mathcal{E}(k)$$
$$= (k+1)(k+2)s \left( f(x_{k+1}) - f(x_k) \right) + 2(k+2)s \left( f(x_{k+1}) - f(x^\star) \right)$$
$$+ \frac{1}{2} \Big\langle 2(x_{k+1} - x_k) + (k+2)\sqrt{s} v_{k+1} - (k+1)\sqrt{s} v_k,$$
$$\qquad 2(x_{k+1} + x_k - 2x^\star) + (k+2)\sqrt{s} v_{k+1} + (k+1)\sqrt{s} v_k \Big\rangle$$
$$= (k+1)(k+2)s \left( f(x_{k+1}) - f(x_k) \right) + 2(k+2)s \left( f(x_{k+1}) - f(x^\star) \right)$$
$$- \Big\langle (k+1)s \nabla f(x_{k+1}), 2(x_{k+1} - x^\star) + (k+2)\sqrt{s} v_{k+1} \Big\rangle$$
$$- \frac{1}{2}(k+1)^2 s^2 \left\| \nabla f(x_{k+1}) \right\|^2.$$

Now, we hope to utilize the basic inequality for $f \in \mathcal{F}_L^1(\mathbb{R}^n)$ to make the right side of equality no more than zero. Based on the following inequalities:

$$\begin{cases} f(x_{k+1}) - f(x_k) \le \langle \nabla f(x_{k+1}), x_{k+1} - x_k \rangle \\ f(x_{k+1}) - f(x^\star) \le \langle \nabla f(x_{k+1}), x_{k+1} - x^\star \rangle - \dfrac{1}{2L} \left\| \nabla f(x_{k+1}) \right\|^2, \end{cases}$$

we obtain:

$$\mathcal{E}(k+1) - \mathcal{E}(k) \le 2s \left( f(x_{k+1}) - f(x^\star) \right) - \frac{(k+1)s}{L} \left\| \nabla f(x_{k+1}) \right\|^2 - \frac{1}{2}(k+1)s^2 \left\| \nabla f(x_{k+1}) \right\|^2.$$

Although the negative term includes the multiplier $k$ and $k^2$, we cannot guarantee that the right-hand-side of the inequality is nonpositive.

Here, in contrast to the subtle discrete construction in Su et al. [2016], we point out that the standard numerical discretization of low-resolution ODE (2.2) cannot maintain the convergence rate from the continuous-time ODE, due the presence of numerical error.