[Reviews · NeurIPS 2019]

Reviewer 1



Update: increased score due to concerns being addressed in author response - Originality That integrating Hamiltonian-type dynamics with symplectic integration achieves better results does not sound surprising. However, formally proving acceleration for standard strongly convex and convex models is by no means an easy task. To the best of my knowledge, this is first work to theoretically show symplectic integrators can achieve acceleration by integrating a specific type of ODE (i.e. the high-resolution polyak heavy-ball or NAG limiting ODE). Related works are sufficiently cited. - Quality Overall, the submission is technically sound, mostly supplying full proofs for theorems and lemmas. However, there are slight technical issues that need to be addressed which I mention in section 5 of the review. - Clarity The clarify is fair in general. I understand that condensing a supposedly 30+ pages theory paper into 8 pages is a daunting task, but I feel the the paper can be greatly improved in this aspect. See section 5 of the review for details. - Significance See section 1 of the review.

Reviewer 2



This paper looks at the ways to derive accelerated methods for smooth convex optimization by discretizing certain ordinary differential equations (ODEs). Specifically, the paper focuses on high-resolution ODEs corresponding to Polyak's heavy ball algorithm and to two variants of Nesterov's accelerated gradient (NAG) algorithm. High-resolution ODEs have been studied recently by Shi et al. 2018, and their main difference from the ODEs in previous work (eg. Su et al. 2016) is that high-resolution ODEs include higher-order step-size terms. The paper considers explicit, implicit, and symplectic discretization schemes applied to these three ODEs. The main finding is that high-resolution ODEs with symplectic discretization schemes yield accelerated discrete-time algorithms. For the low-resolution ODEs that this paper considers, only implicit discretization yields accelerated iteration complexity. For the high-resolution ODEs considered, explicit discretization is unstable, while symplectic and implicit schemes are stable. However, symplectic methods are generally superior to implicit methods as implicit methods require solving a nontrivial subproblem in each iteration. The authors use a Lyapunov analysis to prove their convergence rates. The strong part of this paper is that it gives a clear presentation of how the different discretization schemes considered yield different results for the ODEs considered. The proofs are also fairly clear. The paper notes an interesting property of symplectic discretization methods that seems to be relevant to this line of work. The weak part of this paper is that while it demonstrates that symplectic methods work for the ODEs considered, it doesn't give much intuition as to why one would expect the method to work, which limits how useful this is in informing future research. Essentially, the only analytic contributions are the idea to use symplectic discretization schemes on higher-order ODEs and the choice of Lyapunov functions (I don't know how non-trivial the Lyapunov functions are -- they seem like they could be standard based on previous work). The actual derivation of the ODEs and analysis seems straightforward from the above. Overall, I think this paper is a weak accept. I think it has a nice idea and demonstrates an interesting phenomenon. However, like previous papers in this line of work, this paper is somewhat unclear on how these techniques might be applied elsewhere or what intuition even suggests that the symplectic method should work. Seeing as the purported goal of this paper is to give a more systematic method for developing new accelerated algorithms, it would be nice if the answer were more than find the right high-resolution ODE and try symplectic discretization to see if it works.'' ------- Post Rebuttal ------- I read the other reviews and the rebuttal. I think the paper would benefit from including the comments on symplectic methods that the authors mention in the rebuttal.

Reviewer 3



The paper is well written and well organised. It is theoretically heavy, although that is necessary given the subject matter. I think this paper will be of wide interest to the community.

[Author Response · NeurIPS 2019]

We thank all reviewers for your very useful comments on our paper. Please find our responses to each reviewer below.

**To Reviewer #2:** Thanks for the thorough reading! Regarding your comments on improvements, please find our response below.

1. We will provide more intuition for high-resolution ODEs in Section 2. At a high level, high-resolution ODEs are finer approximations to NAGs that also consider $O(\sqrt{s})$-order terms in contrast to low-resolution ODEs which only consider $O(1)$-order terms. These finer approximations allow us to investigate more refined aspects of the dynamics.

2. Note that Direct RK admits the convergence rate $O(k^{-2s/(s+1)})$ which can be faster than gradient descent but is slower the optimal $O(k^{-2})$ rate. We will clarify and discuss this explicitly in the final version.

3. Thanks for your suggestion. We will modify our paper accordingly.

4. The gradient correction term is $\frac{k}{k+1} \cdot s\left(\nabla f(x_k) - \nabla f(x_{k-1})\right)$ for NAG-C and $\frac{1-\sqrt{\mu s}}{1+\sqrt{\mu s}} \cdot s\left(\nabla f(x_k) - \nabla f(x_{k-1})\right)$ for NAG-SC. Note the effect of the gradient correction term can be ascertained from High-resolution ODEs but not from low-resolution ODEs. We will add a clarification in this regard in the final version.

5. The two algorithms are the standard NAG-SC and the symplectic discretization of high-resolution ODEs.

6. We will modify it accordingly.

7. We will add more detail and description. Thanks for the suggestion.

8. Note that the coefficient function in Theorem B.1 (Theorem B.2) is a continuous function of $s$. Therefore the maximum value of this function in Theorem 3.1 (Theorem 3.2) is achieved on the closed interval.

9. Yes, that is a typo. Thanks for pointing out.

**To Reviewer #3:** Thanks for asking about intuition for symplectic methods. We provide some basic intuition here; we will incorporate these comments in the final version.

One explanation for the superiority on the symplectic method comes from Physics. The symplectic structure is an essential property of Hamiltonian systems, capturing aspects of their geometry that are relevant to invariances of the dynamical flow. For numerical simulation of the Hamiltonian system, the symplectic method involves a forward discretization step and a backward discretization step; together these steps exploit the symplectic geometry such that low-error discretization errors cancel. In contrast, the explicit Euler scheme has two forward discretizations and the implicit Euler two backward discretization; in neither case do the low-order discretization errors cancel. The cancellation of error terms means that larger step sizes can be taken while retaining stability; this is the core of the connection to acceleration.

Consider the linear Hamiltonian system $H(x,y) = \frac{1}{2}(x^2 + y^2)$ whose trajectories are a family of closed curves. After some calculations, one can show that the explicit Euler scheme will diverge to infinity and the implicit Euler scheme will converge to zero. However, the symplectic Euler scheme can guarantee the numerical solution is a closed curve with properly chosen step size.

**To Reviewer #4:** Thanks for your very positive comments as well as helpful suggestions! We will consider adding numerical experiments in the final version.

[Meta-Review · NeurIPS 2019]

This paper presents new technical results connecting discretization of ODEs to accelerated optimization. However the results are a bit niche. Pros: The main finding is that high-resolution ODEs with symplectic discretization schemes yield accelerated discrete-time algorithms. For the low-resolution ODEs that this paper considers, only implicit discretization yields accelerated iteration complexity. However, symplectic methods are generally superior to implicit methods as implicit methods require solving a nontrivial subproblem in each iteration. Cons: Like previous papers in this line of work, this paper is somewhat unclear on how these techniques might be applied elsewhere or what intuition even suggests that the symplectic method should work.